

# Quantifying uncertainties due to chemistry modeling - evaluation of tropospheric composition simulations in the CAMS model

Vincent Huijnen[1], Andrea Pozzer[2], Joaquim Arteta[3], Guy Brasseur[4], Idir Bouarar[4], Simon Chabrillat[6], Yves Christophe[6], Thierno Doumbia[3], Johannes Flemming[5], Jonathan Guth[3], Béatrice Josse[3], Vlassis A. Karydis[2,7],Virginie Marécal[3], Sophie Pelletier[3]

[1]Royal Netherlands Meteorological Institute, De Bilt, The Netherlands
[2]Max Planck Institute for Chemistry, Mainz, Germany
[3]Centre National de Recherches Météorologiques Université de Toulouse, Météo-France, CNRS, Toulouse, France
[4]Max Planck Institute for Meteorology, Hamburg, Germany
[5]European Centre for Medium Range Weather Forecast, Reading, RG2 9AX, UK
[6]Royal Belgian Institute for Space Aeronomy, BIRA-IASB, 1080 Brussels, Belgium
[7]Forschungszentrum Jülich, Inst Energy & Climate Res IEK-8, D-52425 Jülich, Germany

*Correspondence to*: (Vincent.Huijnen@knmi.nl)

**Abstract.**

We report on an evaluation of tropospheric ozone and its precursor gases in three atmospheric chemistry versions as implemented in ECMWF's Integrated Forecasting System (IFS), referred to as IFS(CB05BASCOE), IFS(MOZART) and IFS(MOCAGE). While the model versions were forced with the same overall meteorology, emissions, transport and deposition schemes, they vary largely in their parameterizations describing atmospheric chemistry, including the organics degradation, heterogeneous chemistry and photolysis, as well as chemical solver. The model results from the three chemistry versions are compared against a range of aircraft field campaigns, ozone sondes and satellite observations, which provides quantification of the overall model uncertainty driven by the chemistry parameterizations. We find that they produce similar patterns and magnitudes for carbon monoxide (CO) and ozone ($O_3$), as well as a range of non-methane hydrocarbons (NMHCs), with averaged differences for $O_3$ (CO) within 10% (20%) throughout the troposphere. Most of the divergence in the magnitude of NMHCs can be explained by differences in OH concentrations, which can reach up to 50% particularly at high latitudes. Also comparatively large discrepancies between model versions exist for $NO_2$, $SO_2$ and $HNO_3$, which are strongly influenced by secondary chemical production and loss. Other, common biases in CO and NMHCs are mainly attributed to uncertainties in their emissions. This configuration of having various chemistry versions within IFS provides a quantification of uncertainties induced by chemistry modeling in the main CAMS global trace gas products beyond those that are constrained by data-assimilation.



## 1. Introduction

The analysis and forecasting capabilities of trace gases are key objectives of the European Copernicus Atmosphere
Monitoring Service (CAMS), in order to provide operational information on the state of the atmosphere. This service relies
on a combination of satellite observations with state-of-the-art atmospheric composition modelling (Flemming et al., 2017).
For that purpose, ECMWF's numerical weather prediction (NWP) system, the Integrated Forecasting System (IFS), contains
modules for describing atmospheric composition, including aerosols (Morcrette et al., 2009; Benedetti et al., 2009),
greenhouse gases (Agustí-Panareda, et al. 2016; Engelen et al., 2009), and reactive gases (Flemming et al., 2015).

Having atmospheric chemistry available within the IFS allows the use of detailed meteorological parameters to drive the fate
of constituents, as well as its capabilities to constrain trace gas concentrations through assimilation of satellite retrievals.
Furthermore, having atmospheric chemistry as an integral element of the IFS enables to study feedback processes between
atmospheric chemistry and other parts of the earth system, such as the impact of ozone in the radiation scheme on
temperature and the provision of trace gases as precursors for aerosol.

Other examples where chemistry modules have been implemented in general circulation models (GCM) for NWP
applications have been, for instance, GEM-AQ (Kaminski et al., 2008; Struzewska et al., 2015), GEMS-BACH (de
Grandpré et al., 2009; Robichaud et al., 2010), the Met Office's Unified Model (Morgenstern et al., 2009, O'Connor et al.,
2014), and, on a regional scale, WRF-Chem (Powers et al., 2017).

The chemistry module that is currently used operationally in the CAMS service originates from the chemistry transport
model TM5 (Huijnen et al., 2010). The chemistry module is based on a modified version of CB05 tropospheric chemistry
(Williams et al., 2013), while stratospheric ozone is modelled using a linear ozone scheme (Cariolle and Deque 1986,
Cariolle and Teyssèdre, 2007). This version, referred to as IFS(CB05), is used in a range of applications, such as for the
CAMS operational analyses and forecasts of atmospheric composition (http://atmosphere.copernicus.eu), and for the
generation of reanalyses: the CAMS Interim Reanalysis (Flemming et al., 2017) and the CAMS Reanalysis (Inness et al.,
2018). Furthermore, this module is used in modelling studies, e.g., to analyse extreme fire events (Huijnen et al., 2016a;
Nechita-Banda et al., 2018), to study the correlation between tropospheric composition with ENSO conditions (Inness et al.,
2015). It has also contributed to model intercomparison studies such as arctic pollution (Emmons et al., 2015), HTAP (e.g.
Huang et al., 2017) and AQMEII (Im et al., 2018).

Other chemistry versions have also been implemented in the IFS, where each version has its choice regarding the gas phase
chemical mechanism, computation of photolysis rates, definition of cloud and heterogeneous reactions, and solver specifics.
This enables flexibility in the choice of the atmospheric chemistry component in the global CAMS system. A model version
which contains the extension of the CB05 scheme with a comprehensive stratospheric chemistry originating from the
Belgian Assimilation System for Chemical ObsErvations (BASCOE, Skachko et al., 2016) has been developed (Huijnen et
al., 2016b). Furthermore, in predecessors of the current system, the MOZART (Kinnison et al., 2007) and MOCAGE




(Bousserez et al., 2007) chemistry transport models had also been coupled with IFS (Flemming et al., 2009) and, afterwards, their chemistry modules technically integrated into the IFS (Flemming et al., 2015). Only recently, three fully functioning systems have been prepared, as are presented here, based on CB05BASCOE, MOZART and MOCAGE chemistry.

Many studies such as HTAP and AQMEII (Galmarini et al., 2017) try to explore the uncertainties of global chemistry modelling through changing emissions. But in such multi-model assessments also meteorological model parametrisations,
such as advection, deposition or vertical diffusion vary (e.g. Emmons et al., 2015; Huang et al., 2017; Im et al., 2018). While such a multi-model approach is appropriate to define the overall uncertainty, it makes it hard to isolate the impact of the differences in the chemistry parameterizations. In this work we study the model spread caused by three chemistry modules that are fully independent, in an otherwise identical configuration for describing meteorology, transport, emissions and deposition. This endeavour intends to provide insights in the uncertainty induced purely by the simulation of chemistry and
as such complements the many model intercomparison studies that try to explore other sources of uncertainty in global atmospheric modelling.

The central application area of tropospheric chemistry analyses and forecasts in the IFS are to provide a global coverage of the current state of atmospheric composition, along with its long-term trends (Inness et al., 2018). These are intensively used as boundary conditions to regional models (Marécal et al., 2015). Uncertainty information is relevant to CAMS users of
global chemistry forecasts, in particular for the trace gases that are not, or poorly, constrained by observations, such as the non-methane hydrocarbons (NMHCs) and reactive nitrogen species. Therefore we focus here not only on the model ability to represent tropospheric ozone ($O_3$) and carbon monoxide (CO), but also include evaluations of the NMHC's, nitrogen dioxide ($NO_2$), nitric acid ($HNO_3$) and sulfur dioxide ($SO_2$).

In this study, we rely on various sets of observations. Comparatively dense in-situ observation networks exist to measure
surface and tropospheric CO and $O_3$, which is further expanded by satellite retrievals for CO and $NO_2$ columns. Observations from aircraft campaigns form a crucial source of information on atmospheric composition, particularly for the NMHC's, and have been used in the past in various modelling efforts and intercomparison studies (e.g. Pozzer et al., 2007; Emmons et al., 2015). Even though all model versions considered here contain both parameterizations for tropospheric and stratospheric chemistry, we limit ourselves to evaluating differences in the tropospheric composition; evaluation of stratospheric
composition is beyond the scope of this work. It is worth noting that each of the versions are constantly developed further over time, which means that particular aspects of the model performance, and as a consequence inter-model spread, are subject to change depending on model version.

The paper is structured as follows. Section 2 provides a description of the various chemistry schemes implemented in IFS. Section 3 provides an overview of the observational datasets used for model evaluation, while in Section 4 a basic
assessment of model differences for tracers playing a key role in tropospheric ozone is provided. Section 5 contains the evaluation against observations of a full year simulation with the three atmospheric chemistry versions of IFS with focus on



tropospheric chemistry. The paper is concluded with a summary and an outlook in Section 6, where also the recent model evolution in the various versions is briefly described.

## 2. Model description

### 2.1. Chemical mechanisms

The three chemistry schemes implemented in the IFS are described in more detail in the following subsections. A brief analysis of elemental differences is given in Sec. 2.1.4

#### 2.1.1. IFS(CB05BASCOE)

For IFS(CB05BASCOE), a merging approach has been developed where the tropospheric and the stratospheric chemistry
schemes are used side-by-side within IFS (Huijnen et al.,2016b). The tropospheric chemistry in the IFS is based on a modified version of the CB05 mechanism (Yarwood et al., 2005). It adopts a lumping approach for organic species by defining a separate tracer species for specific types of functional groups. Modifications and extensions to this include an explicit treatment of C1 to C3 species as described in Williams et al., (2013), and $SO_2$, di-methyl sulphide (DMS), methyl sulphonic acid (MSA) and ammonia ($NH_3$) (Huijnen et al., 2010). Heterogeneous reactions and photolysis rates in the
troposphere depend on the CAMS aerosol fields. The reaction rates for the troposphere follow the recommendations given in either JPL evaluation 17 (Sander et al., 2011) or Atkinson et al. (2006).

The modified band approach (MBA) is adopted for the online computation of photolysis rates in the troposphere (Williams et al., 2012) and uses 7 absorption bands across the spectral range $202 - 695$ nm, accounting for cloud and aerosol optical properties. At instances of large solar zenith angles (71-85°) a different set of band intervals is used. The complete chemical
mechanism as applied for the troposphere is referred to as 'tc01a', and is extensively documented in Flemming et al. (2015).

For the modelling of atmospheric composition above the tropopause, the chemical scheme and the parameterization for Polar Stratospheric Clouds (PSC) have been taken over from the BASCOE system (Huijnen et al., 2016b), version 'sb14a'. Lookup tables of photolysis rates were computed offline by the TUV package (Madronich and Flocke, 1999) as a function of log-pressure altitude, ozone overhead column and solar zenith angle. Gas-phase and heterogeneous reaction rates are taken
from JPL evaluation 17 (Sander et al., 2011) and JPL evaluation 13 (Sander et al., 2000), respectively.

Both for solving the tropospheric and stratospheric reaction mechanism we use KPP-based four stages, $3^{rd}$ order Rosenbrock solvers (Sandu and Sander, 2006). Photolysis rates for reactions occurring both in the troposphere and stratosphere are merged at the interface, in order to ensure a smooth transition between the two schemes. To distinguish between the tropospheric and stratospheric regime, we use a chemical definition of the tropopause level, where tropospheric grid cells are
defined at $O_3$<200 ppb and CO>40 ppb, for P > 40 hPa. With this definition the associated tropopause pressure ranges in practice approximately between 270 and 50 hPa globally, with the lowest tropopause pressure naturally in the tropics.





### 2.1.2. IFS(MOCAGE)

The MOCAGE chemical scheme   (Bousserez et al, 2007, Lacressonnière et al. 2012) is a merge of reactions of the
tropospheric RACM (Regional Atmospheric Chemistry Mechanism) scheme (Stockwell et al., 1997) with the reactions
relevant to the stratospheric chemistry of REPROBUS (REactive Processes Ruling the Ozone BUdget in the Stratosphere)
(Lefèvre et al., 1994, Lefèvre et al. 1998). It uses a lumping approach for organic trace gas species. The MOCAGE
chemistry has been extended, in particular by the inclusion of the sulphur cycle in the troposphere (Ménégoz et al. 2009) and
PAN photolysis.

The RACMOBUS (RACM-REPROBUS) chemistry scheme implemented in IFS uses 115 species in total, including long-
lived and short-lived species, family groups and a polar stratospheric clouds (PSC) tracer.  A total of  326 thermal reactions
and  53 photolysis  reactions  are  considered  to  model  both  tropospheric  and  stratospheric  gaseous  chemistry.  Nine
heterogeneous reactions are taken into account for the stratosphere and 2 for the troposphere for the aqueous oxidation
reaction of sulfur dioxide into sulfuric acid (Lacressonnière et al., 2012).For photolysis rates, a lookup table of photolysis
rates was computed offline by the TUV package (Madronich and Flocke, 1997, version 5.3.1) as a function of solar zenith
angle, ozone column above each cell, altitude and surface albedo.

### 2.1.3. IFS(MOZART)

The atmospheric chemistry in IFS(MOZART) is based on the MOZART-3 mechanism (Kinnison et al., 2007) and includes
additional  species  and  reactions  from  MOZART-4  (Emmons  et  al.,  2010)  and  further  updates  from  the  Community
Atmosphere Model with interactive chemistry, referred to as CAM4-chem (Lamarque et al., 2012; Tilmes et al., 2016). As
for IFS(CB05BASCOE), the heterogeneous reactions in the troposphere are parameterized based on aerosol surface area
density (SAD) which is derived using the CAMS aerosol fields. The heterogeneous chemistry in the stratosphere accounts
for heterogeneous processes on liquid sulfate aerosols and polar stratospheric clouds following the approach of Considine et
al. (2000).

The photolysis frequencies in wavelengths from 200 to 750 nm are calculated from a look-up table, based on the 4-stream
version of the Stratosphere, Troposphere, Ultraviolet (STUV) radiative transfer model (Madronich et al., 1989). For
wavelengths from 120 nm to 200 nm, the wavelength-dependent cross sections and quantum yields are specified and the
transmission function is calculated explicitly for each wavelength interval. In the case of J(NO) and J($O_2$), detailed
photolysis parameterizations are included online. The current IFS(MOZART) version includes the influence of clouds on
photolysis rates which is parameterized according to Madronich (1987). However, currently it does not account for the
impact of aerosols. A detailed description of the parametrization of photolysis frequencies, absorption cross sections, and
quantum yields is given in Kinnison et al. (2007).




### 2.1.4. Key differences in chemistry modules

An overview of the most important differences in the three chemistry modules described above is given in Table 1. First, there are large differences in the choices made to compile the tropospheric chemistry mechanism. IFS(MOZART) describes the degradation of organic carbon types C1, C2, C3, C4, C5, C7 and C10, together with lumped aromatics, while

IFS(CB05BASCOE) only describes explicit degradation up to C3, with the same reactions as present in IFS(MOZART). Instead, emissions and degradation of higher VOC's in IFS(CB05BASCOE) are lumped to a few tracers. Furthermore, the parameterization of the isoprene and terpenes degradation is simpler in IFS(CB05BASCOE) than in IFS(MOZART). Aromatics are currently not described in IFS(CB05BASCOE), while they are accounted for with simple approaches in IFS(MOZART).

IFS(MOCAGE) describes many more lumped organic species than IFS(CB05BASCOE) and IFS(MOZART), also accounting for the more complex organics beyond C3. Furthermore, IFS(MOCAGE) uses a rather different lumping approach and contains more complexity for different terpene components, and also including aromatics. Such differences are bound to impact the effective degradation of VOCs, and thus ozone production efficiency and oxidation capacity, e.g. Sander et al. (2018).

With respect to the inorganic chemistry, the schemes are mostly similar. Still, IFS(MOCAGE) includes HONO chemistry, which is missing in both IFS(CB05BASCOE) and IFS(MOZART) implementations. Gas-phase sulfur chemistry is mostly similar between IFS(CB05BASCOE) and IFS(MOZART), while IFS(MOCAGE) has some more complexity through considering reactions involving DMSO and $H_2S$.

Heterogeneous reactions of $HO_2$ and $N_2O_5$ on aerosol are included in IFS(CB05BASCOE) and IFS(MOZART), but not in

the IFS(MOCAGE) version considered here. This has only become available in a more recent model version.

Regarding the treatment of photolysis in the troposphere, IFS(CB05BASCOE) applies a modified band approach, where for 7 wavelengths the photolysis rates are computed online, taking into account the scattering and absorption properties of gases, (overhead ozone and oxygen), clouds and aerosol. IFS(MOCAGE) adopts a lookup-table approach, accounting for overhead ozone column, solar zenith angle, surface albedo and altitude, providing photolysis rates for clear-sky conditions. The impact

of cloudiness on photolysis rates is applied online in IFS during the simulation using the parameterisation proposed by Brasseur et al. (1998). IFS(MOZART) applies the lookup-table approach from MOZART-3 (Kinnison et al., 2007), considering overhead ozone column and cloud scattering effects on photolysis rates. Despite such larger differences, an intercomparison of an instantaneous field of photolysis rates showed similar average profiles, with spread in magnitudes in the range of 5% in the tropical free troposphere for important photolysis rates like $jO_3$, $jNO_2$, $jHNO_3$. Locally differences are

larger, associated, amongst others, to different cloud treatment (Hall et al., 2018).

As for the stratospheric chemistry, IFS(CB05BASCOE) contains the largest complexity of the three model versions containing both more species and reactions compared to the other mechanisms.



Different methods are used to solve the reaction mechanism. IFS(CB05BASCOE) applies the Rosenbrock solver,
IFS(MOCAGE) here applies a first-order semi-implicit solver with fixed time steps, and IFS(MOZART) applies the explicit
Euler method for species with long lifetimes (e.g. N$_2$O) and an implicit backward Euler solver for other trace gases with
short lifetimes. Experiments using different solvers for both IFS(CB05BASCOE) and IFS(MOCAGE) have revealed
significant differences, with decreases in tropospheric ozone in the order of up to 20% regionally when replacing a semi-
implicit solver with the Rosenbrock solver. These differences are mostly traced to differences in the N$_2$O$_5$ chemistry
(Cariolle et al., 2017), affecting in turn the NO$_x$ lifetime and hence the ozone production efficiency.

Table 1. Specification of elemental aspects describing the three chemistry versions.

|  | IFS(CB05BASCOE) | IFS(MOCAGE) | IFS(MOZART) |
|---|---|---|---|
| Tropospheric chemistry | Carbon Bond | RACM | CAM4-Chem |
| Stratospheric chemistry | BASCOE | REPROBUS | MOZART3 |
| Number of species | 99 | 115 | 115 |
| Number of thermal reactions | 219 | 326 | 266 |
| Number of photolysis rates | 60 | 53 | 51 |
| Photolysis parameterization | Modified Band (trop) LUT (strat) | LUT | LUT (trop) Explicit transmission function (strat) |
| Number of heterogeneous reactions | 10 | 11 | 11 |
| Solver | 3$^{rd}$ order Rosenbrock | 1$^{st}$ order semi-implicit | Explicit forward and implicit backward Euler |



### 2.2. Emission, deposition and surface boundary conditions

MACCity emissions are used to prescribe the anthropogenic emissions (Granier et al., 2011), where wintertime CO traffic emissions have been scaled up according to Stein et al. (2014). Aircraft NO emissions are 1.8 Tg NO yr$^{-1}$ , following Lamarque et al. (2010).

Monthly specific biogenic emissions originating from MEGAN-MACC (Sindelarova et al., 2014) are adopted, complemented with POET-based oceanic emissions (Granier et al., 2005).

Daily biomass burning emissions are taken from the Global Fire Assimilation System (GFAS) version 1.2, which is based on satellite retrievals of fire radiative power (Kaiser et al., 2012). The actual emission totals used in the simulation for 2011 from anthropogenic, biogenic and natural sources, biomass burning as well as lightning NO are given in Table 2.

Dry deposition velocities in the current configuration were provided as monthly mean values from a simulation using the approach discussed in Michou et al. (2004). To account for the diurnal variation in deposition velocities, a cosine function of the solar zenith angle is adopted with a ±50% variation. Wet scavenging, including in-cloud and below cloud scavenging as well as re-evaporation is treated following Jacob et al. (2000). The reader is referred to Flemming et al., (2015) for further details on dry and wet deposition parameterization.

As described above, the chemistry mechanisms vary particularly in their description of the VOC degradation, with the most explicit treatment described in IFS(MOZ), while IFS(MOCAGE) and IFS(CB05BASCOE) rely on a more extended lumping approach. This has consequences for the description of the various emissions. Still, we have ensured that the total of VOC and aromatics emissions in terms of Tg carbon are essentially the same for the three chemistry schemes.

For CB05BASCOE, the emissions of Parafins (toluene and higher alkane emissions), Olefins (butenes and higher alkenes), and Aldehydes (acetaldehyde and other aldehydes) have been prescribed. Likewise, MOZART applies emissions of BIGALK (Butanes and higher alkanes) and BIGENE (Butenes and higher alkenes). MOCAGE adopts tracers HC3, HC5, and HC8, over which emissions of acetylene, propane, butanes and higher alkanes, esters, methanol and other alcohols are distributed, whereas DIEN contains butenes and higher alkenes emissions.

As for the aromatics, IFS(CB05BASCOE) disregards those, but includes toluene carbon emissions as part of the Parafins. IFS(MOZART) treats additionally a TOLUENE tracer, while IFS(MOCAGE) contains two types of aromatics, designated TOL and XYL. These aromatic emissions are composed from toluene, trimethyl-benzene, xylene and other aromatics.

Methane, N$_2$O and a selection of CFC's are prescribed at the surface as boundary conditions. While for N$_2$O and CFC currently annually and zonally fixed values are assumed (Huijnen et al., 2016b), for CH$_4$ zonally and seasonally varying surface concentrations are adopted based on a climatology derived from NOAA flask observations ranging from 2003 to 2014.



*Table 2. Specification of annual emission totals from anthropogenic, biogenic and natural sources and biomass burning for 2011, in Tg species, for three chemistry versions.*

| Species | Anthropogenic | Biogenic+oceanic | Biomass burning |
|---|---|---|---|
| CO | 602 | 91+20 | 326 |
| $NO^a$ | 71.2+1.8 AC | 11.3+9.2 LiNO | 8.8 |
| HCHO | 3.4 | 4.8 | 4.8 |
| $CH_3OH$ | 2.2 | 127 | 6.7 |
| $C_2H_6$ | 3.3 | 0.3+1.0 | 2.2 |
| $C_2H_5OH$ | 2.2 | 19.3 | 0. |
| $C_2H_4$ | 7.6 | 30+1.4 | 3.9 |
| $C_3H_8$ | 4.0 | 1.3 | 1.2 |
| $C_3H_6$ | 3.5 | 15.2+1.5 | 2.3 |
| $CH_3CHO$ and higher aldehydes | 1.3 | 23.5 | 3.8 |
| $CH_3COCH_3$ | 1.4 | 38 | 1.8 |
| Butanes and higher alkanes | 35. | 0.1 | 2. |
| Butenes and higher alkenes | 4.7 | 3.1 | 1.6 |
| $C_5H_8$ | | 593 | |
| Terpenes | | 95 | |
| $SO_2$ | 97 | 13 | 1. |
| DMS | | 38 | 0.2 |
| $NH_3$ | 43 | 2+8 | 6.5 |

[a]Anthropogenic surface NO emissions (Tg NO) are split according to 90% NO and 10% $NO_2$ emissions.

Additionally, they contain a contribution of 1.8 Tg NO aircraft emissions and 9.2 Tg NO lightning emissions (LiNO).



### 2.3. Model configuration and meteorology

The IFS model versions evaluated here were implemented in IFS cycle 43R1, and are run on a T255 horizontal resolution (~
0.7 degree) with 60 model levels in the vertical up to 0.1 hPa, all excluding chemical data assimilation. The naming
conventions and experiment ID's for the three model runs are specified in Table 3. For brevity we refer to the model runs as
'CBA', 'MOC' and 'MOZ', respectively. A 30 minutes time stepping for the dynamics is applied while meteorology is
nudged towards ERA-Interim. To allow for sufficient model spinup, the initial condition (IC) fields have been generated for
1 July 2010, using as much as possible realistic and consistent fields. For this purpose, tropospheric CO, $O_3$ from the CAMS-
Interim reanalysis (Flemming et al., 2017) have been combined with VOC's from its control run. CFC's, halogens and other
tracers relevant for stratospheric composition originate from the BASCOE reanalysis v05.06, (Skachko et al., 2016), and
have been merged for altitudes below tropopause with model fields from Huijnen et al. (2016b), all specified for 1 July 2010.
For MOZ and MOC, these IC fields have been completed for a few missing VOC's and CFC's using separate MOZART and
MOCAGE climatologies, respectively.

For the evaluation, the model was sampled in the troposphere and lower stratosphere (i.e. the lowest 40 model levels) every
three hours, to have a full coverage of the daily cycle. These are used to compute monthly to yearly averages. Standard
deviations are computed to represent the model variability for a specified range in time and space.

*Table 3. Specifications of the experiments in evaluated.*

| name | Short name | expID | Color-coding |
|------|------------|-------|--------------|
| IFS(CB05BASCOE) | CBA | a028 | **red** |
| IFS(MOCAGE) | MOC | b0l8 | **blue** |
| IFS(MOZART) | MOZ | b0w3 | **Green** |


### 3. Observational datasets

### 3.1 Aircraft Measurements

Aircraft measurements of trace gas composition from a database produced by Emmons et al. (2000) were used for
evaluation. Although these measurements cover only limited time periods, they provide valuable information about the
vertical distribution of the analyzed trace gases. The database is formed by data from a number of aircraft campaigns,
gridded onto global maps, forming data composites of chemical species important for tropospheric ozone photochemistry.
These are used to create observation-based climatologies (Emmons et al. 2000). Here we use measurements from ozone, CO,
$CH_2O$, $C_2H_6$, $C_2H_4$, methyl hydroperoxide ($CH_3OOH$), $NO_2$, nitric acid ($HNO_3$), and sulphur dioxide ($SO_2$). Note that the
field campaigns used in this evaluation have been extended including also data observed after the year 2000, such as the



TOPSE and TRACE-P campaigns. The geographical distribution of the aircraft campaigns and their area coverage are shown
in Figure 1.

Although the specific field campaign data is in theory representative for the specific year, the averaging of large number of
measurements over space and time partly solves the problem of interannual variability, and therefore these data can be
considered as a climatology. Pozzer et al. (2009) showed that the correlation between model results and these observations

would vary less than 5% if model results 5 years apart were used. Finally these data summaries are useful for providing a
picture of the global distributions of NMHCs and nitrogen-containing trace gases.

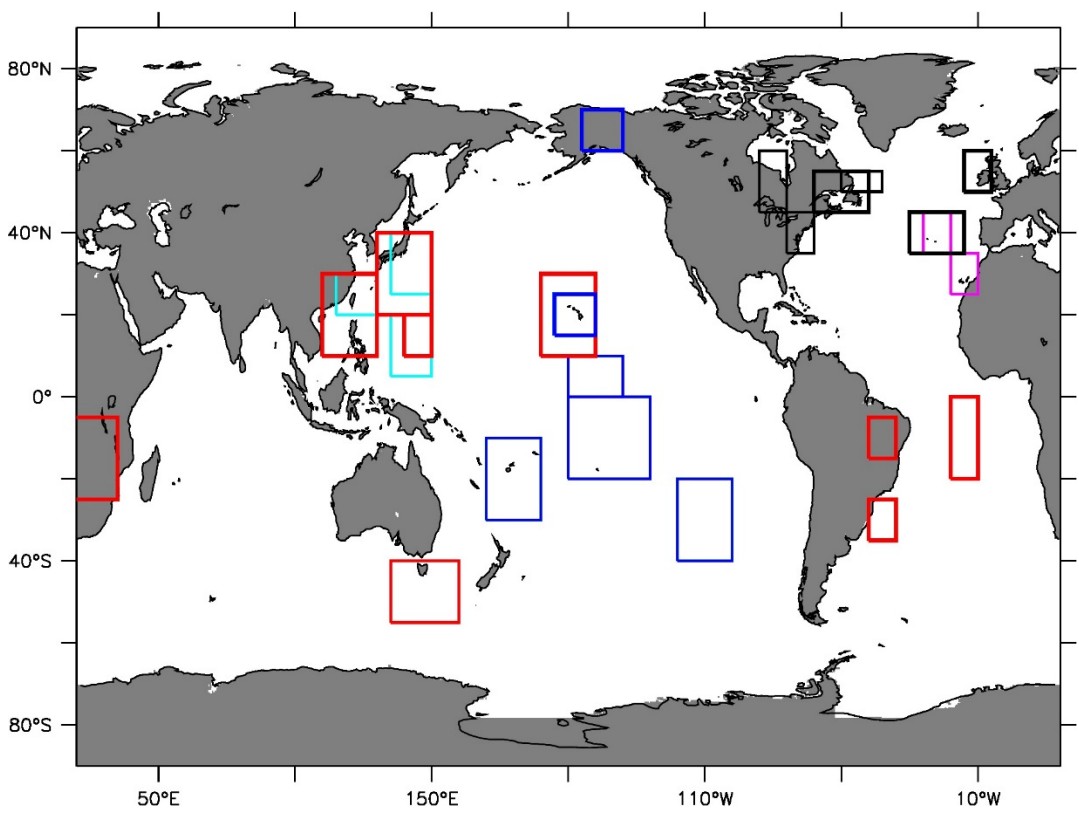


**Figure 1**: Geographical distribution of the aircraft campaigns presented by Emmons et al. (2000). Each field campaign is
represented by a different color. Further information on the campaigns is found in Emmons et al. (2000).



### 3.2. In situ observations

In-situ observations for CO are used from the World Data Centre for Reactive Gases (WDCRG), the data repository and archive for reactive gases of the World Meteorological Organisation's (WMO) Global Atmosphere Watch (GAW) programme. The uncertainty of the CO observations is estimated to be in the order of 1–3 ppm (Novelli et al., 2003). Tropospheric ozone was evaluated using sonde measurement data available from the World Ozone and Ultraviolet Radiation Data Center (WOUDC, http://woudc.org), further expanded with observations from the Network for the Detection of Atmospheric Composition Change (NDACC) network. About 50 individual stations covering various worldwide regions are taken into account for the evaluation over the Arctic, Northern Hemisphere mid-latitudes, tropics, Southern Hemisphere mid-latitudes and Antarctic. The precision of ozone sonde observations in the troposphere is on the order of -7 to 17% (Komhyr et al., 1995; Steinbrecht et al., 1998), while larger errors are found in the presence of steep gradients and where the ozone amount is low.

### 3.3. Satellite observations

MOPITT (Measurements of Pollution in the Troposphere) V7 CO column observations (Deeter et al., 2017) are used to evaluate the CO total columns. The MOPITT instrument is a multi-channel Thermal InfraRed (TIR) and Near InfraRed (NIR) instrument operating onboard of the Terra satellite. The total column CO product is based on the integral of the retrieved CO volume mixing ratio profile. A CAM4-chem (Lamarque et al., 2012) based climatology is used to provide the MOPITT *a priori* profiles. For our study we use the TIR-derived CO total column observations, which are provided both over the oceans and over land. Highest CO sensitivities of these MOPITT TIR measurements are in the middle troposphere, around 500 hPa. Sensitivity to the lower troposphere depends on the thermal contrast between the land and lower atmosphere, which is higher during the day than in the night. Therefore, in our study we only use daytime MOPITT TIR observations. Standard deviation of the error in individual pixels for the MOPITT V7-TIR product evaluated against NOAA flask measurements is reported as $0.13 \times 10^{18}$ molec cm$^{-2}$ (Deeter et al., 2017), i.e. in the order of 10% of the observation value. Daily mean model CO columns have been gridded to a 1° x 1° spatial resolution, and for our analysis we applied the MOPITT averaging kernels to the logarithm of the mixing ratio profiles, following Deeter et al. (2012).

OMI retrievals of tropospheric $NO_2$ were taken from the QA4ECV dataset (Boersma et al., 2017). For this evaluation the 3-hourly model output of $NO_2$ was interpolated in time to local overpass of the satellite (13:30h), while pixels with satellite-observed radiance fraction originating from clouds greater than 50% were filtered out. The averaging kernels of the retrievals are taken into account, hence making the evaluation independent of the a priori $NO_2$ profiles used in the retrieval algorithm. Note that by using the averaging kernels the model levels in the free troposphere are given relatively greater weight in the column calculation, which means that errors in the shape of the $NO_2$ profile can contribute to biases in the total column.





## 4. Assessment of inter-model differences

In this section we provide a basic assessment of magnitude and differences in annual and zonal mean concentration fields between the three chemistry versions for a few essential tracers: $O_3$, CO, $NO_x$ (=NO+$NO_2$) and OH. This provides a first insight in the correspondences and differences between chemistry modules and will help to interpret more quantitative

differences seen in the evaluation against observations.

The annual zonal mean $O_3$ mixing ratios (Figure 2, top) show very similar patterns, with overall low values over the southern hemisphere and highest over the Northern Hemisphere (NH) mid-latitudes, associated to the dominating emission patterns. Differences between chemistry versions are in the order of 10%, with MOC showing comparatively the lowest values over the tropical free troposphere and MOZ the highest, over the NH extra-tropics. Differences in tropospheric ozone between

model versions are remarkably small on a global scale.

Likewise, annual zonal mean CO mixing ratios show highest values associated to pollution regions in the tropics and over the NH. Highest values are obtained with CBA, and lowest with MOC, with differences ranging between 10 and 20%, suggesting mainly differences in oxidizing capacity.

Zonal mean $NO_x$ mixing rations, a tracer playing a crucial role in ozone formation, show overall highest values for MOC and

lowest for CBA. MOZ and CBA are overall similar, but MOC is showing higher values over the NH high-latitudes (>60°N) and also at altitudes below 900 hPa in the tropics. This is likely related to the fact that in this version of IFS(MOCAGE) the coupling with the aerosol module has not yet been established, contrary to CBA and MOZ. Additionally, Cariolle et al. (2017) showed limitations of the Semi-Implicit method as used in MOC for resolving $NO_x$ chemistry. Both elements likely contribute to significantly larger tropospheric $NO_x$ lifetimes in MOC compared to CBA and MOZ. In contrast, the $NO_x$

lifetime in IFS(CB05BASCOE) scheme is comparatively short, which is associated to a diagnosed relatively efficient organic nitrate production term from the reaction of $NO_x$ with VOC's in the modified CB05 mechanism compared to other mechanisms, as assessed in a box-modeling configuration (Sander et al., 2018).

Figure 2 also shows the annual, zonal mean concentrations of OH. Overall, the magnitude of OH is largest for MOC and

lowest for CBA, with MOZ in between. The largest differences in absolute terms are found in the tropics, where the concentrations are highest. Nevertheless, in relative terms the largest differences are found in the extra-tropics, particularly over the SH, as can be seen from Figure 3. This figure shows the temporal evolution of the difference between MOC and MOZ simulated daily average OH at 600 hPa. This shows that differences can be up to 50% in daily averages, in particular over the extra-tropics where the absolute values are lower compared to those in the tropics.

Tropospheric $NO_x$ in MOC is comparatively high, suggesting relatively efficient $O_3$ and OH production. On the other hand, the photolysis rates of tropospheric ozone, responsible for the primary production of OH, are very similar (not shown). Therefore the ozone production in MOC must be counter-balanced by a relatively large loss through reaction with OH and $HO_2$ (which are the other major loss terms in the ozone cycle), suggesting a relatively short tropospheric $O_3$ lifetime. Such





differences in oxidation capacity naturally have important implications for understanding differences in the performance of

NMHCs, as discussed in the next sections.

**Figure 2:** Zonal, annual mean O₃, CO, NO$_x$, mixing ratios and OH concentrations in CBA (left), MOZ (middle) and MOC (right).






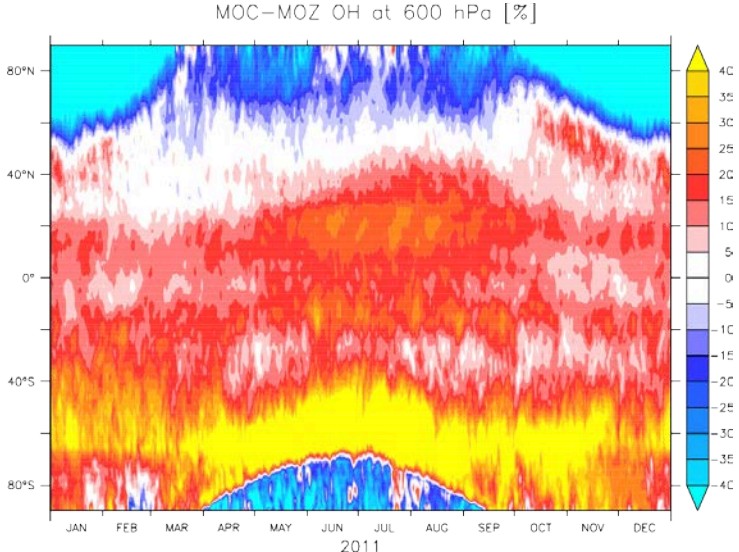

**Figure 3:** Relative differences (in %) of OH daily averaged mixing ratios of simulation MOC with respect to MOZ at 600
hPa.

## 5. Evaluation against observations

In this section we evaluate the model simulations against a range of observations, including ozone sondes, aircraft
measurements, and satellite observations from carbon monoxide and nitrogen dioxide.

Table 4 summarizes the comparison of the various model results with aircraft measurements, described in Sec. 3.1, in terms
of biases and correlation, both unweighted and weighted with uncertainties which are approximated by the root mean square
of model variability and measurement variability, where model variability is represented by the standard deviation from the
averaged output values and measurement variability is by combination of instrumental errors and standard deviation(Jöckel
et al., 2006). With this approach, the measurement locations with high variability have less weight. This allows us to
compare values that are more representative for the average conditions and to eliminate specific episodes that cannot be
expected to be reproduced by the model. According to this analysis, the discrepancies between model results and
measurements are smaller than the uncertainties, if the absolute value of the weighted bias (i.e., in units of the normalised
standard deviation, Table 4) for a specific tracer is less than one. A high weighted correlation in combination with a
weighted bias between [-1,1] indicates that the model is able to reproduce the observed mixing ratios on average. This holds
for all versions for CO, $O_3$, $CH_2O$, $NO_2$, and $HNO_3$, while model versions have more difficulties with $CH_3OOH$. For $SO_2$
CBA is the only model version to deliver a satisfactory bias. For $C_2H_4$ and $C_2H_6$ none of the versions are able to match the
observations to an acceptable degree. The inability of the model versions to reproduce the observed magnitude of $C_2H_6$ and
the vertical distribution of $C_2H_4$, as indicated by the relatively low correlation with all aircraft measurements included in the
database, requires a more detailed analysis. This is investigated in more detail in the next sections.




*Table 4. Summary of the Bias and correlation coefficients (R) of model results versus all available aircraft observations, also weighted with relative uncertainties. Bias = model results minus observations. Bias$^a$ and R$^{2\,a}$ is in pmol/mol, (except for CO and O$_3$). Bias$^b$ and R$^{2\,b}$ are given in standard deviation units.*

| Tracer | N. obs | CBA | | | | MOC | | | | MOZ | | | |
|---|---|---|---|---|---|---|---|---|---|---|---|---|---|
| | | Bias$^a$ | Bias$^b$ | R$^{2\,a}$ | R$^{2\,b}$ | Bias$^a$ | Bias$^b$ | R$^{2\,a}$ | R$^{2\,b}$ | Bias$^a$ | Bias$^b$ | R$^{2\,a}$ | R$^{2\,b}$ |
| O$_3$* | 506 | 10.6 | 0.32 | 0.57 | 0.60 | 10.1 | 0.40 | 0.59 | 0.65 | 15.9 | 0.71 | 0.58 | 0.71 |
| CO* | 457 | -2.11 | 0.35 | 0.22 | 0.88 | -14.7 | -0.43 | 0.21 | 0.86 | -14.1 | -0.38 | 0.21 | 0.89 |
| CH$_2$O | 213 | -13.7 | -0.11 | 0.63 | 0.76 | 20.1 | 0.31 | 0.67 | 0.72 | 24.3 | 0.26 | 0.70 | 0.80 |
| CH$_3$OOH | 366 | -46.5 | -0.47 | 0.58 | 0.93 | 51.4 | 0.15 | 0.69 | 0.88 | -114 | -0.92 | 0.74 | 0.96 |
| C$_2$H$_4$ | 454 | -6.28 | -4.80 | 0.58 | 0.39 | -5.35 | -2.78 | 0.54 | 0.03 | -4.02 | -13.8 | 0.54 | 0.06 |
| C$_2$H$_6$ | 473 | -505 | -3.18 | 0.50 | 0.81 | -562 | -3.90 | 0.44 | 0.77 | -524 | -3.50 | 0.46 | 0.79 |
| NO$_2$ | 264 | 6.09 | 0.24 | 0.34 | 0.98 | 49.9 | 0.39 | 0.27 | 0.98 | 8.89 | -0.24 | 0.33 | 0.99 |
| HNO$_3$ | 416 | -45.3 | -0.32 | 0.40 | 0.86 | -14.3 | -0.12 | 0.38 | 0.83 | -49.7 | -0.34 | 0.43 | 0.90 |
| SO$_2$ | 350 | -17.0 | -0.63 | 0.18 | 0.87 | -48.7 | -2.25 | 0.16 | 0.95 | -31.2 | -1.20 | 0.49 | 0.88 |

*\* Bias$^a$ for CO and O$_3$ is given in units nmol/mol.*

### 5.1. Ozone (O$_3$)

Figure 4 compare tropospheric O$_3$ profiles simulated by the three model versions with ozone sonde observations for six
different regions over the four seasons. Figure 5 shows annually averaged model biases for various latitude bands and for altitude ranges 900-700hPa, 700-500hPa and 500-300hPa (Figure 5). Overall the three chemistry versions deliver similar performance, reproducing the regionally averaged variability in O$_3$ observations, with various biases depending on the season, region and altitude range. Typically, the model versions tend to simulate lower O$_3$ mixing ratios in the SH mid and high-latitudes compared to sonde observations, and higher in the tropics. Over the Arctic, Western Europe, Eastern US and
Tropics, MOZ simulates too high O$_3$ concentrations at all altitudes and for all seasons except in June-July-August (JJA), with average positive biases ranged from 1 to 12 ppbv in the free troposphere. Here it is worth mentioning that recent updates to





reaction probabilities and aerosol radius assumptions in the heterogeneous chemistry module in IFS(MOZART) significantly improved $O_3$ concentrations particularly in the NH.

MOC shows positive biases over the NH mid latitudes during winter and spring and negative biases during Arctic winter in
the lower troposphere (<700hPa) as well as in the 700-300hPa range in summer. CBA simulates $O_3$ mixing ratios that are generally in close agreement with observations over the Arctic and NH mid-latitudes, but negative biases up to 10 ppbv are obtained in the Arctic upper troposphere (500-300hPa) during winter time (Figure 4, top panel). All three model versions are consistently too high close to the surface (> 800hPa) over the tropics for all seasons, but particularly during December-January-February (DJF). Over the Antarctic and, to a lesser extent, the Southern Hemisphere (SH) mid-latitudes all three
model versions underestimate $O_3$, with negative biases up to 10 ppbv for a large part of the year. However, it should be noted that in the SH regions this evaluation is less representative because there are very few observations.


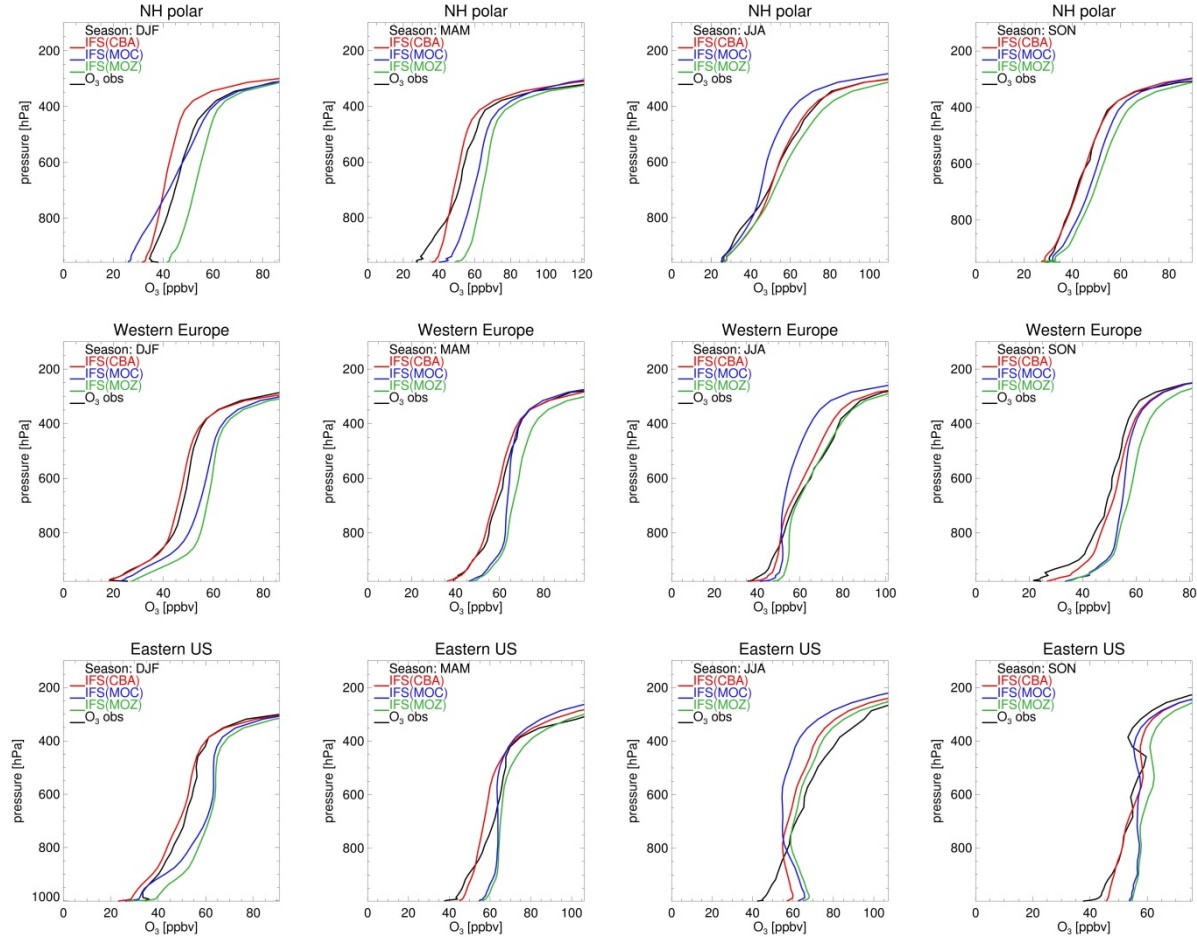






**Figure 4**: Tropospheric ozone profiles of volume mixing ratios (ppbv) against by model versions CBA (red), MOC (blue) and MOZ (green) against sondes (black) over six different regions: from top row to bottom row, NH-Polar [90°N-60°N], Western Europe [45°N-54°N; 0°E-23°E], Eastern US [32°N-45°N; 90°W-65°W], Tropics [30°N-30°S], SH mid-latitudes [30°S-60°S] and Antarctic [60°S-90°S], averaged over four seasons (from left to right: December-January-February, March-April-May, June-July-August, September-October-November).





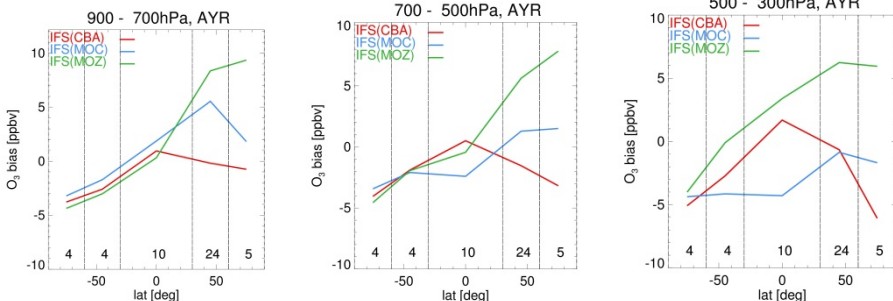

**Figure 5:** Mean of all model bias values against ozone sondes as function of latitude for various altitude ranges, averaged over the full year. Same color codes as in the previous figure. The numbers in each latitude range indicate the amount of stations that contribute to these statistics.

The summary analysis of the average performance of IFS simulations to model tropospheric $O_3$ as compared to WOUDC sondes as provided in Figure 5 shows annually and zonally averaged biases within ±10 ppbv, which is also in line with the $O_3$ bias statistics against the aircraft climatology as provided in Table 4. Nevertheless, this evaluation highlights common discrepancies between model versions and observations, such as the negative bias over Antarctica and positive bias below 700 hPa for tropical stations (see also Figure 4), suggesting biases in common parameterizations such as transport, emissions and deposition.

Figure 6 shows an evaluation of $O_3$ profiles against sondes at selected individual WOUDC sites representative of the Arctic (Ny-Alesund), NH mid-latitudes (Lindenberg), Tropics (Hong Kong, Nairobi), SH mid-latitudes (Lauder) and Antarctic (Neumayer) for DJF and JJA seasons in 2011. We note generally similar biases as compared to those for the regional averages, even though local conditions play a larger role explaining the different performance statistics for these stations. Overall, the evaluation at individual station provides reasonable agreement between model simulations and sondes.




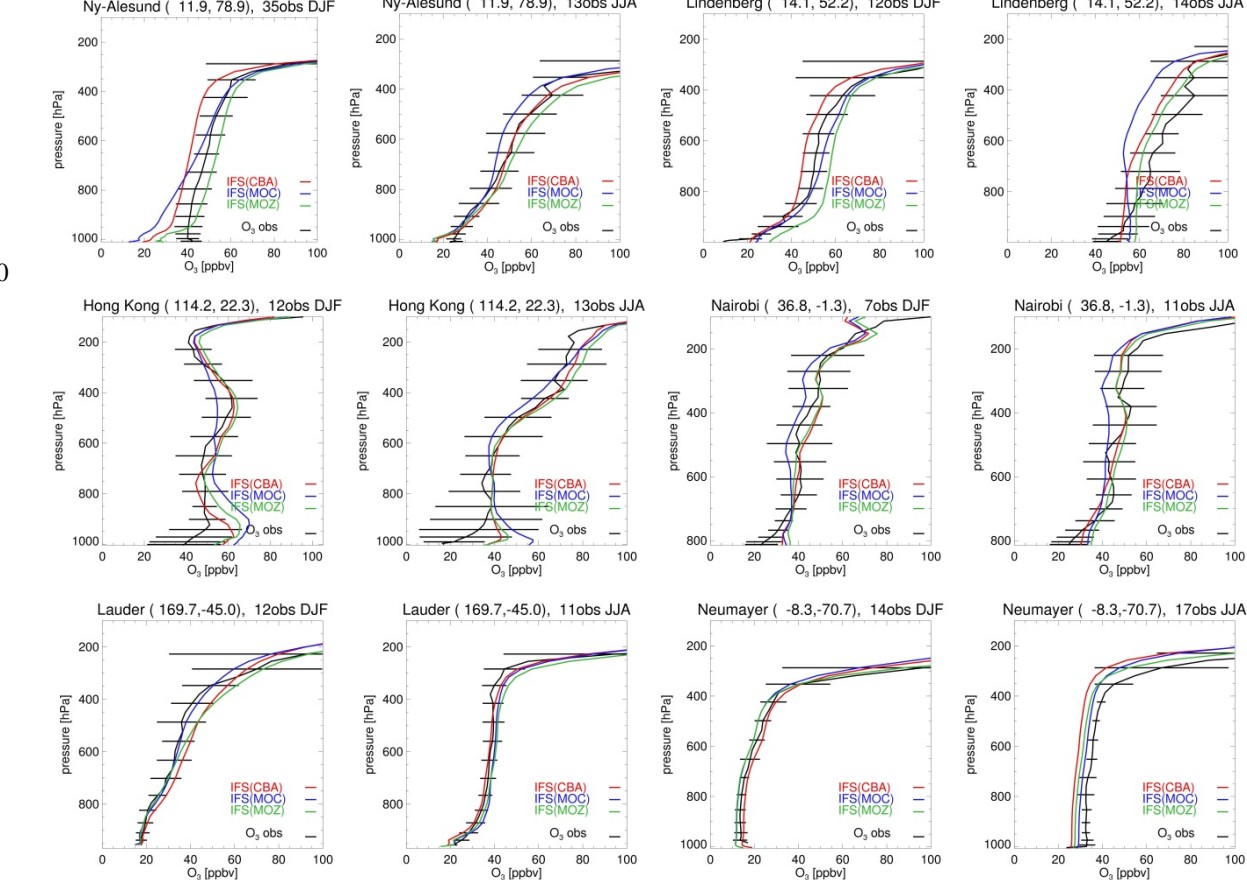

**Figure 6**: Mean profiles of ozone volume mixing ratios during DJF and JJA at selected individual stations. Error bars represent the 1σ spread in the seasonal mean observations.


### 5.2. Carbon Monoxide (CO)

Carbon monoxide is a key tracer for tropospheric chemistry, as a marker of biomass burning and anthropogenic pollution, and provides the most important sink for OH. Approximately half of the CO burden is directly emitted, and the rest formed through degradation of methane and other VOC's. Hence, a correct simulation of this tracer is very important for studies of

atmospheric oxidants. Considering the use of the same emissions and $CH_4$ surface conditions, differences in CO concentrations are essentially caused by differences in chemistry.

Figures 7 and 8 show the monthly mean evaluation against MOPITT total CO columns for April and August 2011. Whereas generally the model versions show good agreement with the observations in terms of their spatial patterns, persistent seasonal biases remain, such as the low bias over the NH during April (further analysed in, e.g., Stein et al., 2014), as well as

a low bias over Eurasia during August. For all three chemistry versions the patterns of enhanced CO in the tropics,



associated to biomass burning, are generally well captured, as well the magnitude of CO columns over the SH. Looking at differences between model versions, CBA shows overall the highest magnitudes, implying a smaller negative bias over the NH particularly during April, while this simultaneously results into an emerging positive bias in the tropics.


**Figure 7**: MOPITT CO total column retrieval for April 2011 (top left) and simulated by IFS(CBA) (top right), IFS(MOZ) (bottom left) and IFS(MOC) (bottom right).








**Figure 8**: MOPITT CO total column retrieval for August 2011 (top left) and simulated by IFS(CBA) (top right), IFS(MOZ) (bottom left) and IFS(MOC) (bottom right).



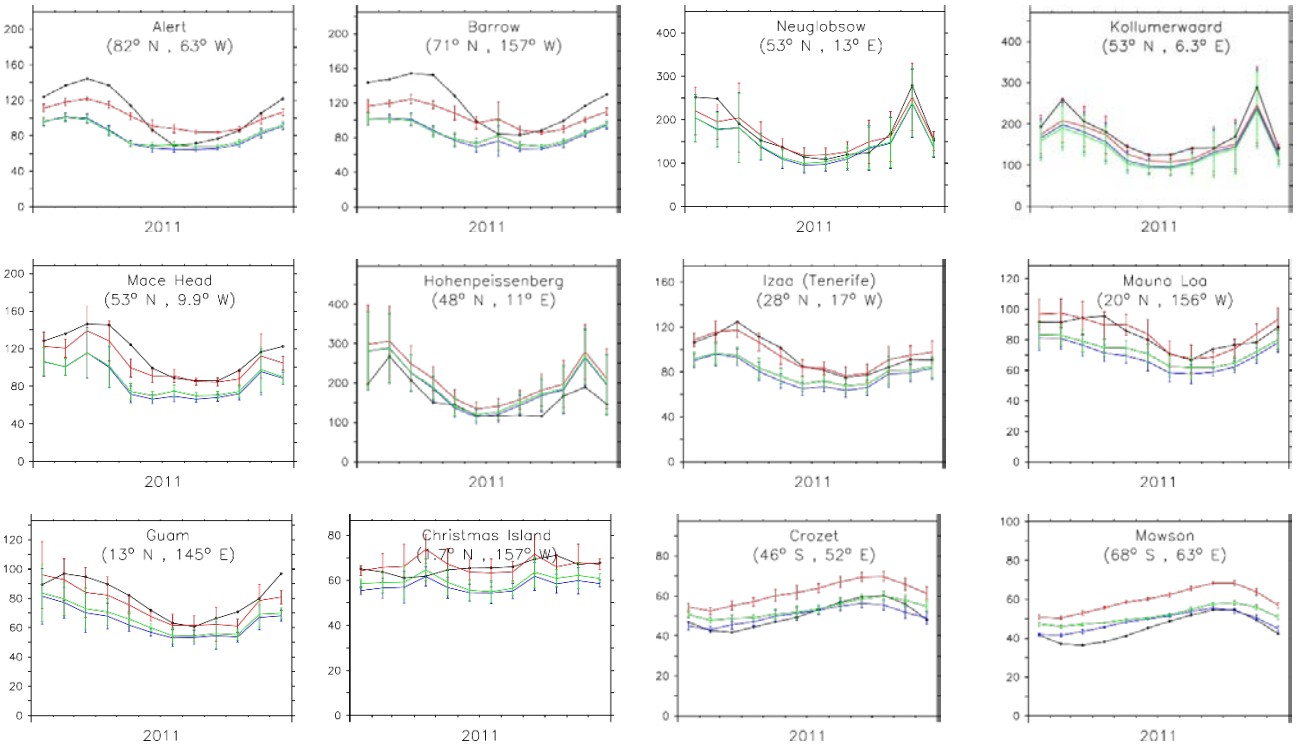

**Figure 9:** Comparison of simulated (red, blue and green are model results from CBA, MOC and MOZ, respectively) and observed (black) CO mixing ratios in nmol/mol at the surface. The bars represent one-standard deviation of the monthly average for the location of the station.

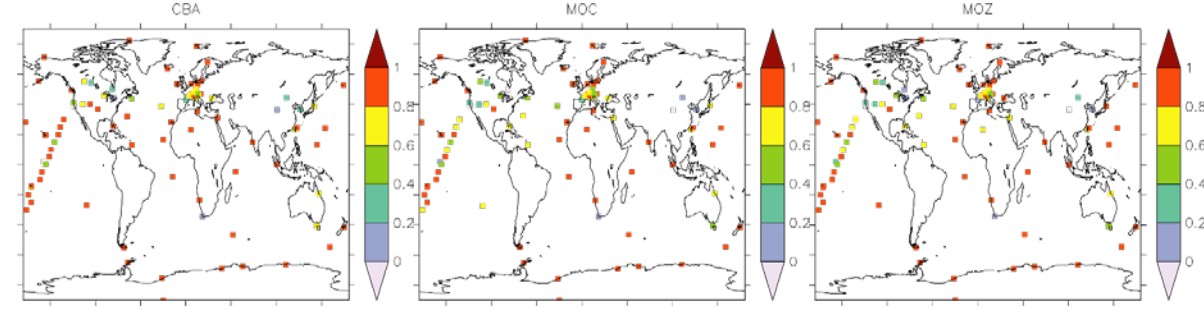

**Figure 10:** Temporal correlation between monthly mean surface CO by observations (GAW network) and by the model simulations (left: CBA, middle: MOC, right: MOZ).

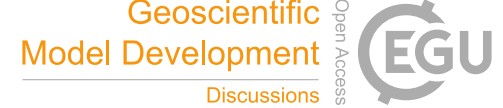



In Figure 9 the annual cycle at selected GAW stations are shown while in Figure 10 additionally shows the corresponding temporal correlation between the simulated monthly mean CO for all stations. Even though the phae and amplitude of the annual cycle are well reproduced by the model versions at several locations (e.g., Mauna Loa, Hawaii), the concentrations tend to be overestimated in the Southern Hemisphere, particularly by CBA, and to a lesser extent by the other chemistry

versions, and underestimated over the remote Northern Hemisphere. This points to sensitivities due to the applied chemistry scheme associated to differences in OH (see also Sec. 4). A possible over-estimation of CO over the tropics and southern hemisphere could relate to uncertainties in the biogenic emissions (Sindelarova et al., 2014).

The correlations (in terms of $R^2$) of monthly mean time series against GAW stations are mostly above 0.8. Particularly over Antarctica the correlation is very high with $R^2 \approx 0.9$, indicating that indeed the main processes controlling the CO abundance

are well represented by the model. Nevertheless, at locations between 40°N and 60°N the correlation is lower. These regions are strongly influenced by local chemistry and emissions, including industry and biomass burning. Clearly, the seasonal cycle is not optimally reproduced in Northern America (Canada regions) by any of the three chemistry versions, indicating that uncertainties in regional emissions, such as boreal biomass burning, could be responsible for these disagreements.

Compared to aircraft observations (see Figure 11), the three model versions produce similar CO mixing ratio vertical profiles, with differences among them typically within the range of 10%. The biomass burning plumes are reproduced consistently (see Figure 11, TRACE-A, West Africa coast), and all three models compare well with observations both for background conditions in the Northern Hemisphere (SONEX, Ireland) and highly polluted condition (PEM-West-B, China Coast).

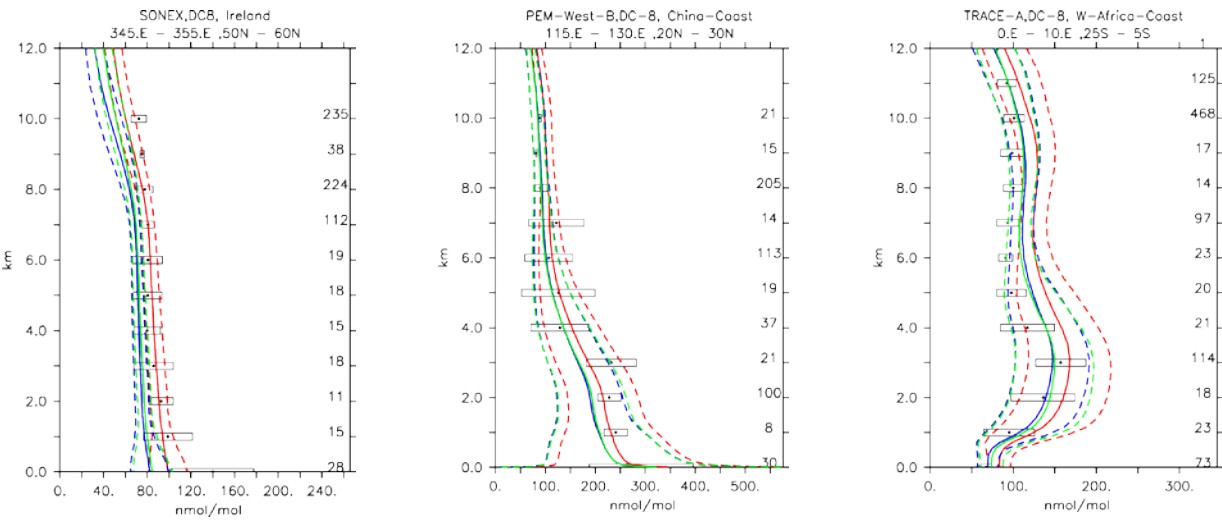


**Figure 11:** Comparison of simulated CO vertical profiles by using the CBA (red solid line), MOC (blue solid line) and MOZ (green solid line) chemistry versions against aircraft data (black dots). Also shown are the modeled (dashed lines) and measured (black rectangular) standard deviations. The numbers on the right vertical axis indicate the number of available measurements.



### 5.3. Formaldehyde (CH$_2$O)

Formaldehyde is important as the most ubiquitous carbonyl compounds in the atmosphere (Fortems-Cheiney et al., 2012). It is mainly formed through the oxidation of methane, isoprene and other VOC's such as methanol (Jacob et al., 2005), while its oxidation and photolysis is responsible for about half of the source of CO in the atmosphere. A good agreement of the simulations with the observations can be seen from Figure 12, where the vertical profile from selected aircraft observations and model simulations are shown. Also from Table 4 it is clear that all the three model versions do reproduce formaldehyde accurately. The weighted bias always well below 1 standard deviation unit (i.e. -0.11, 0.31 and 0.26 for CBA, MOC and MOZ, respectively), indicating that the simulations are well within the statistical uncertainties.

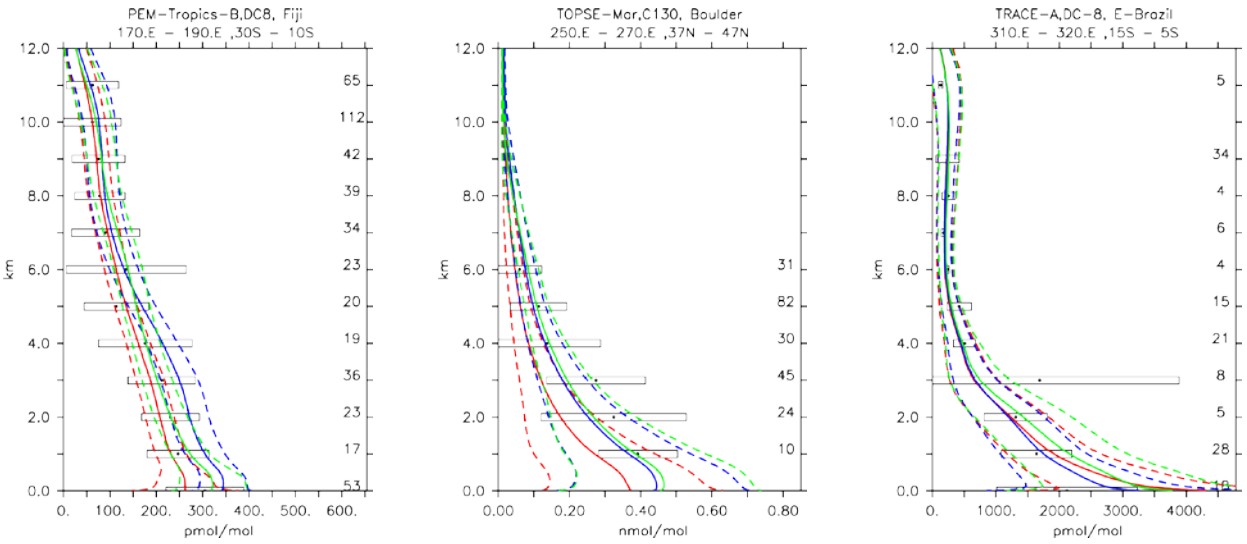

**Figure 12:** Comparison simulated CH$_2$O vertical profiles by using the CBA (red), MOC (blue) and MOZ (green) chemistry versions against aircraft data (black), see also caption of Figure 11.



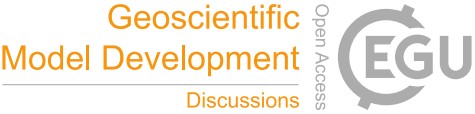

### 5.4. Ethene ($C_2H_4$)

Ethene is the smallest alkene which is primarily emitted from biogenic sources. In our configuration, biogenic $C_2H_4$ emissions are 30 Tg yr$^{-1}$, which appears at the upper end of such emission estimates as reported by Toon et al. (2018). The rest of the emissions are attributed to incomplete combustion from biomass burning or anthropogenic sources.

The three chemical mechanisms produce mostly very similar mixing ratios of $C_2H_4$. Nevertheless, as indicated by the bias (Table 4), which ranges between -2 and -14 in standard deviation units, as well as the weighted correlations, the model versions have difficulties in simulating $C_2H_4$. The vertical profiles (see Figure 13) are strongly biased (e.g., SONEX, Newfoundland and PEM-Tropics-A, Tahiti), with positive biases occurring at the surface and negative in the free troposphere. In remote regions and at higher altitudes, where the direct influence of emissions is lower, the model is at the lower end of the range of observations, with frequent underestimates (see Figure 13 PEM-Tropics-A, Christmas Island). This was already observed in other studies (e.g. Pozzer et al. 2007), implying that the chemistry of this tracer is not well understood. As the underestimation appears to be ubiquitously distributed this suggests that $C_2H_4$ decomposition is too strong, or that the model versions miss chemical production term (e.g., Sander et al., 2018).

Furthermore, interesting is the comparatively large difference present between the simulations at high latitudes (e.g. SONEX, Newfoundland), where the largest relative differences in modelled OH have been found, (see also Sec. 4), illustrating the importance of OH for explaining inter-model differences.

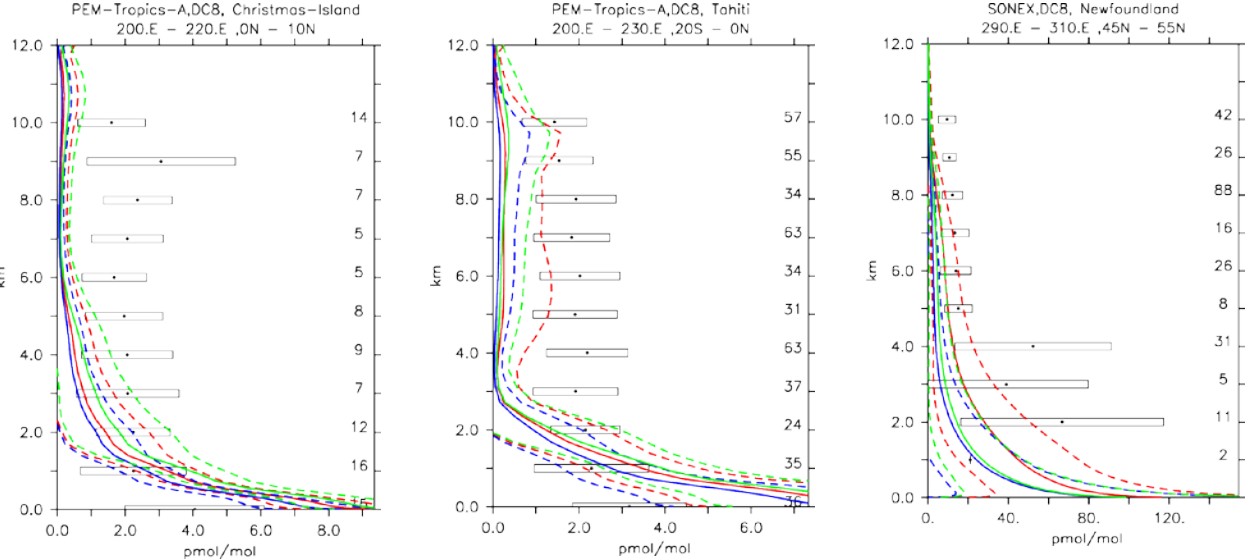

**Figure 13:** Comparison simulated $C_2H_4$ vertical profiles by using the CBA (red), MOC (blue) and MOZ (green) chemistry versions against aircraft data (black), see also caption of Figure 11.



### 5.5. Ethane (C$_2$H$_6$)

Ethane (C$_2$H$_6$) is the lightest trace gas of the family of alkanes and has an atmospheric lifetime of about two months. Ethane emissions are primarily of anthropogenic nature, and have seen a relatively strong decrease since the 1980s (Aydin et al., 2014). Nevertheless, since 2009 an increase in C$_2$H$_6$ concentrations has been observed, believed to be associated to recent

increases in CH$_4$ fossil fuel extraction activities (Hausmann et al., 2016, Monks et al., 2018).

Compared to aircraft observations, all three model versions significantly underestimate the C$_2$H$_6$ observed mixing ratios at all locations and ubiquitously (see Figure 14). A particularly strong underestimation is found in the Northern Hemisphere, where most of the observations are located (e.g. the SONEX campaign over Ireland). A strong negative bias was also reported in the overall statistics (Table 4), even though, contrarily to C$_2$H$_4$, the correlation showed acceptable values for all

versions (R$^2$>0.7). These findings can well be explained by an underestimation of the MACCity-based C$_2$H$_6$ emissions, which are at least a factor two lower than the corresponding estimates of 12-17 Tg yr$^{-1}$ reported in the literature (Monks et al., 2018, Aydin et al., 2014, Emmons et al., 2015; and Folberth et al., 2006). On the other hand, the comparison with the TRACE-A field campaign, which covered long-range transport of biomass burning plumes, shows a reasonable agreement in the lower troposphere (1-4 km), i.e. at the location of the biomass plume, suggesting appropriate biomass burning emissions.

Still a considerable underestimation is present in the upper troposphere, probably due to the missing background concentration.

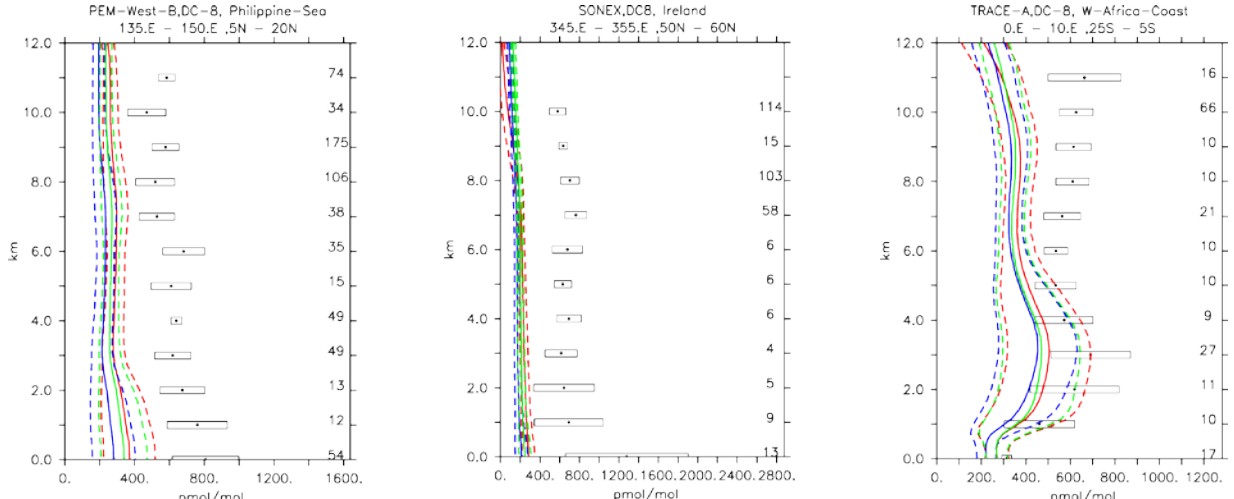

**Figure 14:** Comparison simulated C$_2$H$_6$ vertical profiles by using the CBA (red), MOC (blue) and MOZ (green) chemistry versions against aircraft data (black), see also caption of Figure 11.




### 5.6. Nitrogen Dioxide (NO₂)

Nitrogen dioxide is a trace gas difficult to compare with in-situ observations, due to its photochemical balance with nitrogen oxide. Considering its strong diurnal cycle, due to the fast photolysis rate, here only daytime values have been used to construct the averages, because the observations from the various field campaigns were equally conducted in daylight
conditions. Figure 15 shows the strong variability in daytime NO₂ values, both in the measurements and in the simulations. In general the MOC simulation shows the highest concentration of NO₂ in different locations, particularly over source regions (see Figure 15, TRACE-P, Japan and TOPSE-Feb, Boulder), with MOZ and CBA being more similar. This is in line with the analysis given in Sec. 4. Outside the source regions the secondary processes (such as its equilibrium with HNO₃, see also next section) have larger influences hence the model and observation profiles of NO₂ show even stronger variability and
larger differences (see Figure 15, TOPSE-May, Thule). Still, in general all the chemical mechanisms are able to reproduce NO₂ within 1 standard deviation (see Table 4), even though the unweighted mean bias for MOC is significantly higher than for CBA and MOZ.

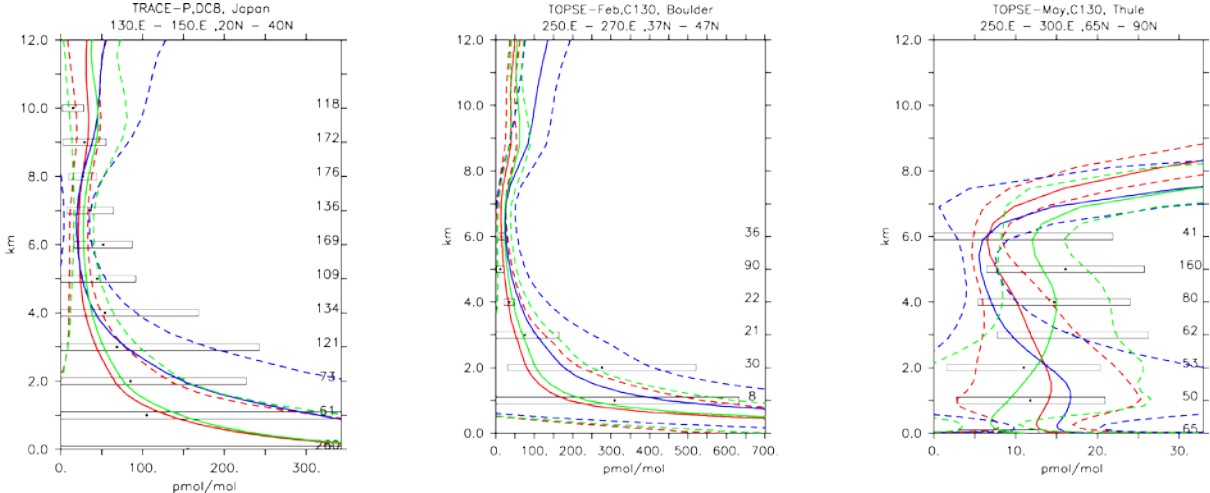

**Figure 15:** Comparison of daytime NO₂ vertical profiles simulated by CBA (red), MOC (blue) and MOZ (green) chemistry versions against aircraft data (black), see also caption of Figure 11.

Figures 16 and 17 evaluate tropospheric NO₂ using the OMI satellite observations. The simulations deliver generally appropriate distributions with a correct extent of the regions with high pollution, as largely dictated by the emission patterns.
Nevertheless, a general underestimation of NO₂ over West Africa in April, and Central Africa and South America in August is found, suggesting uncertainties associated to the modelling of biomass burning emissions.
Another interesting finding is a relatively strong negative bias over the Eurasian and North American continents in April for CBA, stronger than modelled in MOZ and MOC. In contrast, particularly MOC, but also MOZ over-estimates NO₂ over the



comparatively clean North Atlantic and North Pacific oceans in April. This all suggests a relatively short $NO_x$ lifetime in
CBA compared to MOZ and MOC, which in turn helps to explain the lower $O_3$ over the NH-mid latitude regions as
modelled with CBA (see Figure 5). In August the differences in tropospheric $NO_2$ between the three model versions are
smaller than in April.

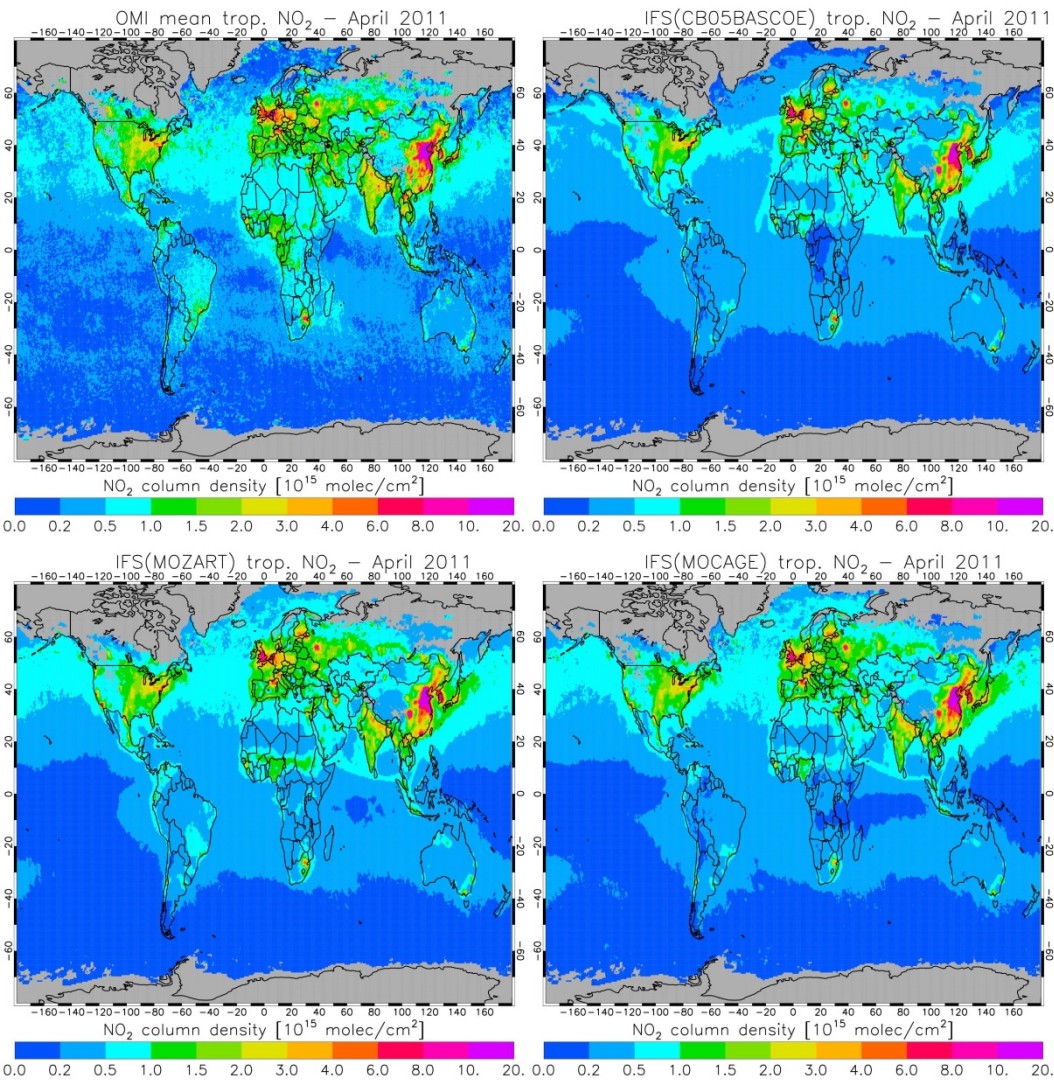


**Figure 16:** Tropospheric $NO_2$ from OMI satellite retrievals from the QA4ECV product for April 2011, along with collocated
model values.





**Figure 17:** Tropospheric NO$_2$ from OMI satellite retrievals from the QA4ECV product for August 2011, and the model biases with respect to this.

### 5.7. Nitric Acid (HNO$_3$)

Compared to the trace gases previously analysed, Nitric acid is not primary emitted but is purely photochemically formed in the atmosphere. It has a very high solubility and therefore tends to be scavenged by precipitation very efficiently, providing an effective sink for the NO$_x$ family. Furthermore, it can act as a precursor for nitrate aerosols (Bian et al., 2017). HNO$_3$ concentrations are therefore expected to show amongst the largest variation between the simulations, as the production and





the sink terms can largely differ due to uncertainties in the parameterizations. In Figure 18, the model results are compared
with selected aircraft measurements. Although all three models tend to reproduce $HNO_3$ in a statistically similar way, over
the lower troposphere and up to 2 km height MOC tends to result in higher $HNO_3$ concentrations compared to the other two
chemical mechanisms and measurements. This is also reflected by overall the lowest negative biases in Table 4. While MOC
performs better at higher altitudes, in a biomass burning plume (e.g. TRACE-A, Figure 18), it also overestimates the
production of $HNO_3$ or underestimates its sinks. Over polluted regions (Figure 18, TRACE-P, Japan), all models tend to
perform well but in remote areas (Figure 18, TOPSE, Churchill) the discrepancies between the models increase with MOC
delivering twice more $HNO_3$ than the other two model versions. Nevertheless, as the variability of the observations is very
large, all the model versions still fall within the range of uncertainties of the observations. The discrepancies between the
model versions can be mainly attributed to differences in $NO_x$ lifetimes and nitrate aerosol formation.


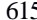

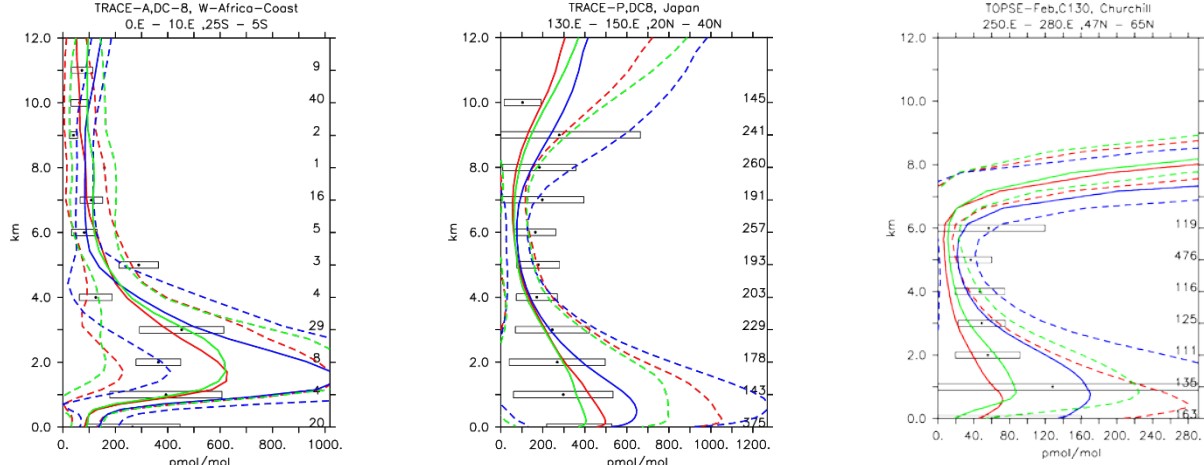

**Figure 18**: Comparison simulated $HNO_3$ vertical profiles by using the CBA (red), MOC (blue) and MOZ (green) chemistry
versions against aircraft data (black), see also caption of Figure 11.

**5.8.  Sulfur Dioxide ($SO_2$)**

Similar to $HNO_3$, $SO_2$ is also strongly influenced by wet deposition due to its high solubility. Furthermore, $SO_2$ is primarily
emitted and converted to sulfuric acid ($H_2SO_4$) both by gas phase and aqueous phase oxidation, an essential process for the
production of new sulfate aerosol particles. Considering the complexity of the processes that control the $SO_2$ fate in the
atmosphere, a large variability is expected for this tracer. The evaluation of $SO_2$ shows that among the three chemistry
versions, CBA produces always the highest $SO_2$ mixing ratios, whereas MOC produce the lowest, and MOZ lies always in
between. Nevertheless, all three mechanisms tend to underpredict $SO_2$ mixing ratios (see Table 4) compared to the aircraft




observations (see Figure 19). Notwithstanding significant uncertainties regarding $SO_2$ emissions, the simulated mixing ratios over polluted regions seem to reproduce the observed values (Figure 19, Trace-P, China and Japan). CBA presents the best comparison with aircraft observations, as can be seen in Figure 19 for the TOPSE aircraft measurements. Also from Table 4,

only CBA delivers a normalized weighted bias within [-1, 1] for $SO_2$, while for the other model versions these are below -1 (-2.25 and -1.20 for MOC and MOZ, respectively).

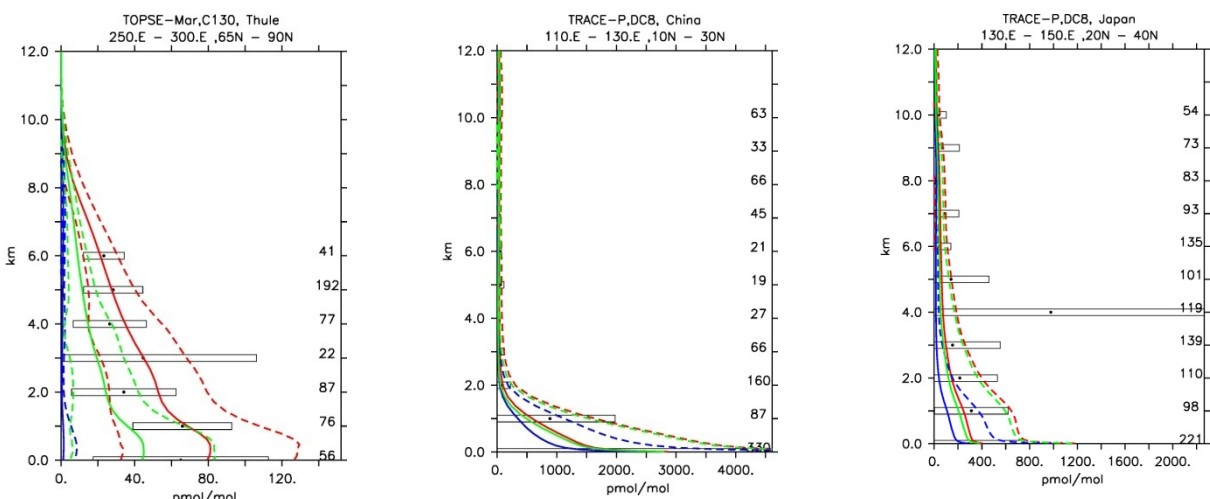

**Figure 19:** Comparison simulated $SO_2$ vertical profiles by using the CBA (red), MOC (blue) and MOZ (green) chemistry versions against aircraft data (black), see also caption of Figure 11.

**Conclusions**

We have reported on an extended evaluation of tropospheric trace gases as modelled in three largely independent chemistry configurations to describe ozone chemistry, as implemented in ECMWF's Integrated Forecasting System. These configurations are based on IFS(CB05BASCOE), IFS(MOZART) and IFS(MOCAGE) chemistry versions. While the model versions were forced with the same overall emissions and adopt the same parameterizations for transport and dry and wet

deposition, they largely vary in their parameterizations describing atmospheric chemistry. In particular their VOC degradation, treatment of heterogeneous chemistry and photolysis, and the adopted chemical solver vary strongly across model versions. Therefore this evaluation provides a quantification of the overall model uncertainties in the CAMS system for global reactive gases which are due to these chemistry parameterizations, as compared to other common uncertainties such as emissions or transport processes.

Overall the three chemistry versions implemented in the IFS produce similar patterns and magnitudes for CO, $O_3$, $CH_2O$, $C_2H_4$ and $C_2H_6$. For instance, the averaged differences for $O_3$ (CO) is within 10% (20%) throughout the troposphere, which is in line with larger model intercomparison studies reported in literature (Emmons et al., 2015; Huang et al., 2017). Except for $C_2H_6$ and $C_2H_4$, all these trace gases are also well reproduced by the various model versions, with an uncertainties-





weighted bias always well within one standard deviation when compared to aircraft observations. Nevertheless the daily average OH levels may vary by up to 50% between the different simulations, particularly at high latitudes where absolute values are smaller. This may explain the larger model spread seen for $C_2H_4$. Comparatively large discrepancies between model versions exist for $NO_2$, $SO_2$ and $HNO_3$, because they are strongly influenced by parameterized processes such as photolysis, heterogeneous chemistry and conversion to aerosol through gas-phase and aqueous phase oxidation. For instance IFS(MOCAGE) tends to predict significantly higher $NO_x$ and $HNO_3$ concentrations in the lower troposphere compared to the other two chemistry versions.

The comparison of the model simulations of NMHCs against a selection of aircraft observations reveals two major issues. First, the evaluation shows that large uncertainties remain in current and widely used emission estimates. For instance, the MACCity ethane emissions are likely under-estimated by at least a factor 2 (Hausmann et al., 2016; Monks et al., 2018) and were shown to lead to significantly lower $C_2H_6$ concentrations compared to the aircraft observations. Secondly, as has been shown before (Pozzer et al., 2007), the significantly lower $C_2H_4$ levels at high altitudes compared to measurements, even though $C_2H_4$ emissions appear in the right order of magnitude, suggest that the $C_2H_4$ chemistry is not well described. Other issues to constrain tropospheric ozone chemistry, as revealed from this assessment, are the model spread in $NO_2$, and its biases against observations. To handle the various discrepancies discussed here, several promising updates are being introduced in the three chemistry versions of IFS, specifically:

- Coupling of the heterogeneous reactions in the troposphere with CAMS-aerosol in IFS(MOCAGE),
- Implementations of more accurate solvers for atmospheric chemistry based on Rosenbrock (Sandu and Sander, 2006) or alternatively ASIS (Cariolle et al., 2017) in IFS(MOCAGE),
- Revisions in atmospheric chemistry scheme in IFS(MOZART) by revising assumptions in the heterogeneous chemistry, expending the complexity of the scheme with additional species, detailed aromatic speciation instead of lumped TOLUENE, and updated reaction products following recent developments in CAM-Chem,
- Update to the look up table for photolysis rate determination in IFS(MOZART),
- Updates of the reaction rate coefficients in any of the chemistry schemes to follow latest recommendations from IUPAC or JPL.

An update of the emission inventories is also foreseen for the near future. All these updates should tend to narrow the spread between the three model versions, and bring them closer to observations. This suggests that the present estimates of uncertainties in atmospheric chemistry parameterizations are on the conservative side. Still, the diversity of chemistry versions will be useful to provide a quantification of uncertainties in key CAMS products due to the chemistry module, as compared to other sources of uncertainties.





## Author contributions

VH designed the study, contributed in the evaluations against sondes and satellite retrievals and wrote large parts of the manuscript. VH, SC, YC and JF developed the IFS(CB05BASCOE) chemistry module, VM, JA, TD, JG, BJ and SP developed the IFS(MOCAGE) chemistry module, IB and GB contributed to the development of the IFS(MOZART) chemistry module, AP and VK performed the evaluation against aircraft observations, and contributed to the writing.

## Data Availability

The model simulation data as used in this work can be obtained upon request from the corresponding author.

## Acknowledgements

We acknowledge funding from the Copernicus Atmosphere Monitoring Service (CAMS), which is funded by the European Union's Copernicus Programme. We are grateful to the World Ozone and Ultraviolet Radiation Data Centre (WOUDC) for providing ozone sonde observations. We thank the Global Atmospheric Watch programme for the provision of CO surface observations. MOPITT data were obtained from the NASA Langley Research Atmospheric Science Data Centre. We acknowledge the free use of tropospheric $NO_2$ column data from the OMI sensor from the QA4ECV project.

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
