# Peer review of "Quantifying uncertainties due to chemistry modeling - evaluation of tropospheric composition simulations in the CAMS model"

_Geoscientific Model Development, 2018_

## Referee Comment (RC1) · Ordóñez (Referee) · 26 Jan 2019

**Review of Huijnen et al.: Quantifying uncertainties due to chemistry modeling - evaluation of tropospheric composition simulations in the CAMS model.**

The manuscript reports an evaluation of tropospheric reactive gases, including ozone and precursors, in three chemistry versions of the IFS: CB05BASCOE (CBA), MOZART (MOZ) and MOCAGE (MOC). The three configurations use the same meteorology, emissions, transport and deposition schemes. Therefore, the differences in the model results should be basically due to differences in the chemical schemes, namely gas phase chemistry, heterogeneous chemistry, photolysis and chemical solver.

Overall, the manuscript is well-written, the presentation quality is good and the authors give credit to related work. As motivated in the introduction, this work documents the diversity in global model simulations caused by chemistry, unlike many other papers which focus on the impact of emissions or other model parameterizations. This is a useful exercise that falls very well within the scope of the journal.

My only concern is that I miss some in depth discussions about the potential reasons for the differences in the performance of the three model configurations considering what is different in them (i.e. the chemistry). The "Main comments" below include some suggestions on how to improve this. These are not really major comments. I hope that some of them are useful, but I am aware of the complexity involved in comparing different chemistry schemes. There are also some minor and technical corrections which need attention. I will fully support the publication of the manuscript in GMD once these issues have been addressed.

**Main comments**

(1) Section 2.1.4 (Key differences in chemistry modules, including Table 1) provides a relatively detailed overview of the differences in the three chemical mechanisms. All this information is very valuable but not completely helpful. A simple summary comparing the complexity of the different processes in the three configurations could be useful to get a broader picture of the main differences. For instance:

1.1. Complexity of organic chemistry: MOCAGE (more extended lumping) ~ MOZART (more explicit) > CB05BASCOE

1.2. Complexity of tropospheric inorganic chemistry: MOCAGE (e.g. includes HONO) > MOZART ~ CB05BASCOE

1.3. Complexity of tropospheric heterogeneous chemistry (HO2 & N2O5): CB05BASCOE ~ MOZART > MOCAGE.

Table 1 indicates that MOCAGE has as many heterogeneous reactions as the other schemes, but that information is not very helpful. I assume those reactions are only relevant for the conversion of SO2 to H2SO4 in the troposphere (lines 139-140) and for stratospheric chemistry, because on lines 179-180 the authors indicate that "Heterogeneous reactions of HO2 and N2O5 on aerosol … not in the IFS(MOCAGE) version considered here".

1.4. Is there a two-way coupling between MOZART / CB05BASCOE oxidants and CAMS aerosol fields? In other words, are the oxidants passed to the aerosol module to produce aerosols, depleted in that processes and then passed back to the chemistry schemes? Or does the coupling only consist of using CAMS aerosols for heterogeneous chemistry in the two chemistry modules? This information is not clear in the main text and is relevant e.g. to evaluate differences in NOx and HNO3.

1.5. Complexity of photolysis: Is it more complex in CB05BASCOE because it includes the impact of aerosols? Anyway, if there is no easy way to compare complexity of photolysis among the three configurations one can leave this out.

1.6 Complexity of stratospheric chemistry: CB05BASCOE > MOZART ~ MOCAGE

1.7 Chemical solver: CB05BASCOE uses Rosenbrock (differences in NOy chemistry and up to ~20% less O3 regionally). It would be interesting to know if the "differences in the N2O5 chemistry" indicated by the authors (line 199) often result in smaller amounts of the species evaluated here (NO2 and HNO3). If not clear, please leave this out.

Summarising, although I understand how hard it is to compare different chemistry schemes, a new table or some bullet points similar to those indicated above could be used to better see the main differences in the three model configurations. Afterwards, one could elaborate on the potential implications for the representation of the main tracers evaluated here (see next comment).

(2) Some additional comments in different parts of the text could be helpful to try to understand the model differences. As mentioned above, I do not expect the authors to be able to explain absolutely everything, but hope that some of these comments will be useful and easy to include in the text:

2.1 Section 4 (Assessment of inter-model differences). The comparison of NOx annual mixing ratios (Figure 2, lines 330-333) could easily be improved:

"MOZ and CBA are overall similar, but MOC is showing higher values over the NH high-latitudes (>60°N) and also at altitudes below 900 hPa in the tropics". First, a minor correction: I clearly see differences in the lower to mid-troposphere from the tropics to the high latitudes, i.e. not only for latitudes >60°N and below 900 hPa in the tropics.

"This is likely related to the fact that in this version of IFS(MOCAGE) the coupling with the aerosol module has not yet been established, contrary to CBA and MOZ". Do you mean that the lack of heterogeneous hydrolysis of N2O5 in MOC will probably raise the NOx concentrations (because that reaction produces the reservoir species HNO3, which is efficiently removed from the atmosphere by dry and wet deposition, resulting in a removal of NOx)? If so, please clearly indicate this. Otherwise, you should explain which other processes might be involved. For instance, is there aerosol nitrate production, removing NOy from the gas phase, in CBA and MOZ?

2.2 The last paragraph of Section 4 (Assessment of inter-model differences) tries to relate the differences in NOx-O3-OH in the models. Overall, the authors point to faster chemical production of O3 in MOCAGE, counterbalanced by fast loss by reaction with OH and HO2. They also suggest a relatively short lifetime of O3 in MOCAGE and conclude the paragraph with "Such differences in oxidation capacity naturally have important implications for understanding differences in the performance of NMHCs, as discussed in the next sections". A couple of questions:

* The authors are probably right, but does the C-IFS have any diagnostics that can be used to quantitatively compare budgets/lifetimes of O3 due to different processes in the three configurations? That would be useful.

* That paragraph should also include a comment for CO. A few lines above (lines 326-328), the authors compare CO in the three configurations (CBA >> MOZ > MOC), "suggesting mainly differences in oxidizing capacity". I think they could be more specific. I am aware that there is no straightforward comparison of the CO and OH levels in the models, because different VOC degradation schemes will impact the CO mixing ratios (as correctly indicated on first paragraph of Section 5.2). Nevertheless, the authors could relate the higher (lower) CO levels in CBA (MOC) to the lower (higher) OH levels seen for the same models in Figure 2. I will come back to this below (comment on Section 5.2).

2.3. Section 5.1 provides a detailed evaluation of ozone. However, after so many comparisons, it is a bit hard to conclude what we have learnt. A summary of the model performance, even if the reasons for the model differences need to be better investigated, might help.

For instance, I know there are some specific comments for MOZ in the first paragraph of the section (e.g. "MOZ simulates too high O3 concentrations ... Here it is worth mentioning that recent updates to reaction probabilities and aerosol radius assumptions in the heterogeneous chemistry module in IFS(MOZART) significantly improved O3 concentrations particularly in the NH"), but a summary mentioning that at the end would be useful.

Some conclusions might be that modelled ozone is often within the uncertainty of the selected sondes (Figure 6), but that MOZ tends to yield higher O3 levels everywhere in the mid-to-upper troposphere (500-300 hPa) and everywhere in the northern hemisphere (Figure 5). Somewhere one should also mention that this configuration yields the highest biases (15.9 ppb), but at the same time the best weighted correlations, when compared to aircraft data (Table 4). Then, one could say again that the biases have partly been corrected in recent versions with improved heterogeneous chemistry.

Finally, the authors could try to repeat Figure 5 but showing accumulated errors (e.g. RMSE) instead of biases. If that provides any further information to document the performance of the three configurations then it should be included.

2.4. Section 5.2 (Carbon monoxide).

The authors include a useful introduction about all the processes impacting CO levels (emissions and VOC degradation) and then comment several plots comparing models with different datasets (MOPITT, surface, aircraft profiles). Overall, I am rather happy with this section, but again it would be nice to finish it with a paragraph summarising what we have learnt from these comparisons.

For instance, from Figures 7-9 (MOPITT & surface CO) one can see that CBA is the configuration with the highest CO levels, yielding the best comparisons (lowest negative biases) in the northern hemisphere but positive biases in the southern hemisphere (Figure 9). The authors should mention that this may partly be related to the fact that this scheme is the one with the lowest OH levels (Figure 2). I would say this is one of the main outcomes from these comparisons.

Towards the end of this section the authors could indicate whether modelled CO is within a given range for the three schemes (they mention 10% only for the comparisons with aircraft observations, but in the abstract and conclusions they indicate 20%). That range (expressed as a percentage or ppb) could be compared to the overall model error for this species (when compared e.g. with surface observations) to get an idea about the fraction of that error that could be attributed to differences in the chemistry. Here or in the "Conclusions" section one could even point to the need of producing some model diagnostics in the future to understand to what extent the differences in the atmospheric mixing ratios of CO arise from differences in CO degradation (by reaction with OH) or differences in CO production (from VOC degradation) in the chemistry schemes.

2.5. Section 5.3 (formaldehyde). It is nice that the authors use weighted biases to show that the simulations are within the range of uncertainty of the few observations available. The correlations are also quite decent. However, this section is rather short and could probably benefit from some improvements and/or discussions.

Lines 506-508 say: "Formaldehyde is important as the most ubiquitous carbonyl compounds in the atmosphere (Fortems-Cheiney et al., 2012). It is mainly formed through the oxidation of methane, isoprene and other VOC's such as methanol (Jacob et al., 2005), while its oxidation and photolysis is responsible for about half of the source of CO in the atmosphere".

First, a couple of minor comments: You should mention that HCHO is not only secondarily formed but also directly emitted. In the first sentence, do you mean "one the most ubiquitous carbonyl compounds"?

I am not sure that the end of the second sentence is right. There the authors mention that the degradation of HCHO is half the source of CO in the atmosphere. Somewhere else (lines 448-449) they mention that "Approximately half of the CO burden is directly emitted, and the rest formed through degradation of methane and other VOCs". Are the authors implying that most of the CO formed (not emitted) in the atmosphere comes from HCHO? Isn't that too much? Before making that statement, the authors should check some references or compare the production terms of CO in the chemistry schemes. If HCHO is so important in terms of CO production, could one try to relate the model performance for both species? For instance, it is possible to relate the fact that CBA is the simulation with the highest CO mixing ratios (which I previously attributed to the low OH in that model) to the negative HCHO biases found for the same scheme in this section?

2.6. Section 5.4 (Ethene, C2H4).

The authors have mentioned the main sources of this species and have provided some insights into the reasons for the model performance. Most of this section looks good to me, but I have some comments:

* The last paragraph says: "Furthermore, interesting is the comparatively large difference present between the simulations at high latitudes (e.g. SONEX, Newfoundland), where the largest relative differences in modelled OH have been found, (see also Sec. 4), illustrating the importance of OH for explaining inter-model differences". Could one explicitly mention that the higher C2H4 mixing ratios in CBA (red in Figure 13, right) compared to the other two schemes are probably related to the lower OH levels in the same scheme?

On the other hand, I am surprised that CBA is the scheme with the largest negative biases for C2H4 in Table 4 (-6.28 ppt). It does not look like that from Figure 13. Is that because the figure only shows data for some specific campaigns and locations? If so, my previous comment might not be valid.

* Lines 531-532: "As the underestimation appears to be ubiquitously distributed this suggests that C2H4 decomposition is too strong, or that the model versions miss chemical production term (e.g., Sander et al., 2018)". This conclusion looks fine, but they should change "miss chemical production term" to "miss some chemical production terms".

Somewhere in this section it would be worthwhile to briefly mention (without all the explanations given below) that one should not expect to get this species right because model and observations are not really collocated in time. I understand that model data are monthly averages (considering both daytime and night-time data) while observations were probably collected at daylight. In addition, model and aircraft observations are from different years. All this is relevant if we keep in mind all the processes affecting the mixing ratios of this species. Many of those processes have already been mentioned by the authors: terrestrial biogenic emissions (probably with strong diurnal cycle), which should be much larger in Newfoundland (Figure 13 right, high C2H4 mixing ratios) than in Christmas Islands and Tahiti (oceanic environment, Figure 13 left and centre, low C2H4); diurnal cycle of anthropogenic emissions (again source probably much higher in Newfoundland); biomass burning sources (with strong interannual variability); upward lifting and subsequent horizontal transport of plumes (not sure if seen on observations from Figure 13 left for Christmas Islands), and relatively short lifetime, mainly controlled by its reaction with OH (higher levels at daylight than at night-time).

2.7 Section 5.6 (Nitrogen dioxide, NO2).

Overall, this section is fine too, but I miss more detailed explanations and some integration with the rest of the text towards the end of the section. In addition, providing maps with biases would be helpful to see the differences between the model configurations and OMI (this is also indicated in a minor comment).

For instance, around lines 582-586, the authors mention the lower NO2 levels for CBA than for the other two models, over polluted and clean regions in the northern hemisphere. Then, around the same lines they say: "This all suggests a relatively short NOx lifetime in CBA compared to MOZ and MOC, which in turn helps to explain the lower O3 over the NH-mid latitude regions as modelled with CBA (see Figure 5)". A couple of comments on this:

* In Section 2.1.4 the authors mentioned some implications of the use of the Rosenbrock solver (in CBA) for NOx and O3, but I think that is hardly mentioned again in the rest of the text. Shouldn't one mention it here?

* CBA has less NOx, O3 and OH, but more CO, than the other model configurations. Not sure if all these facts are related. Shouldn't this be mentioned somewhere?

2.8. HNO3 (also related to NOx).

There is a good introduction relevant to HNO3 around lines 601-604: "It has a very high solubility and therefore tends to be scavenged by precipitation very efficiently, providing an effective sink for the NOx family. Furthermore, it can act as a precursor for nitrate aerosols (Bian et al., 2017). HNO3 concentrations are therefore expected to show amongst the largest variation between the simulations, as the production and the sink terms can largely differ due to uncertainties in the parameterizations". However, later on, the authors are not very specific about the potential reasons for the model differences they find. They just add some general sentences like these ones:

Lines 612-613 (for HNO3): "The discrepancies between the model versions can be mainly attributed to differences in NOx lifetimes and nitrate aerosol formation"

Lines 653-655 (in the conclusions): "For instance IFS(MOCAGE) tends to predict significantly higher NOx and HNO3 concentrations in the lower troposphere compared to the other two chemistry versions".

Can one relate some of the model differences to the lack of coupling of MOCAGE chemistry with the aerosol module? I assume this is not simple. For instance, the inclusion of heterogeneous loss of N2O5 in CBA and MOZ should remove NOx from the system but form HNO3, which would not explain the high HNO3 in MOC. On the other hand, if there is a two-way coupling between the chemistry and the aerosol modules, CBA and MOZ will provide oxidants (e.g. HNO3) to the aerosol model to form aerosols (e.g. nitrate) and the oxidants will be depleted in that process. This could counteract the production of HNO3 by heterogeneous reactions. Are all these processes taking place in CBA and MOZ but not in MOC?

And is wet deposition done exactly in the same way for the three configurations? Otherwise, it should have an impact on the HNO3 mixing ratios.

**Minor comments**

* line 22: "compared against a range of aircraft field campaigns, ozone sondes and satellite observations". Surface observations should also be mentioned here (even if only used for CO).

* Lines 25-26: "Most of the divergence in the magnitude of NMHCs can be explained by differences in OH concentrations". Why only NMHCs? Shouldn't one mention CO too?

* Chemical names, formulae and abbreviations are not used in a consistent way throughout the text. There are some examples here, but this list is not exhaustive at all (the authors should revise the whole text):

"PAN" should be changed to "peroxyacetyl nitrate (PAN)" on line 135.

"polar stratospheric clouds (PSC)" needs to be mentioned only once (for B05BASCOE). In the description of the other two configurations the authors can simply use "PSC".

It should be "VOCs" and not "VOC's". The same applies to "NMHC's" and "CFC's". Please, also spell out VOC and CFC.

In particular, special attention is required in Section 2.2 (Emissions). There the authors use capitals for names such as "Olefins", "Aldehydes", "Butenes" or "Parafins", but that is not needed.

The names "methane" and "CH4" are used in different parts of the text, but the authors never indicate "methane (CH4)". All names of chemical species should be given together with their formulae the first time they appear in the text.

They authors use the name "acetylene", while the IUPAC name (ethyne, C2H2) might be more appropriate.

When possible they should indicate both the name and the formula for all the species in Table 2 (otherwise, please give both name and formula in the main text). There should be a reference to this table from the very beginning of section 2.2.

* Section 2.2. Please indicate whether the non-anthropogenic emissions (e.g. biogenic VOCs and NO, NO from lightning) are prescribed or interactive.

* The authors have indicated the experiment IDs in Table 3. This and other information they provide will be very useful to ensure traceability. I have also noted that in the "Data Availability" section they say: "The model simulation data as used in this work can be obtained upon request from the corresponding author". Are the model experiments from Table 3 stored on ECMWF's servers and accessible to other users? If that is the case, the data availability section could be rewritten.

* In Section 2.3 (Model configuration and meteorology) the authors mention that "the model was sampled … every three hours. These are used to compute monthly to yearly averages. Standard deviations are computed to represent the model variability for a specified range in time and space". This is fine but, after reading section 3 (Observational datasets), it is not clear how the model output and observations are sampled for many of the comparisons shown in the text. I assume that the data are always collocated in space but not completely collocated in time. For instance, aircraft observations are from different years. Please indicate for all datasets whether they are for the simulated year (i.e. 2011) or for another period. Then indicate whether all comparisons are made using model and observations from the same month (even if the data are from different years).

* Section 3.1 (Aircraft measurements). The authors used the compilation of Emmons et al. (2000) as well as TOPSE and TRACE-P data. If it is not too complicated, they could also indicate the names of all the campaigns used, the period covered and so on. In Figure 1 they indicate that "Each field campaign is represented by a different colour" but they do not mention the names. Later, some of those names appear on the profile plots (Figure 11, 12, …). I simply think it would be helpful to know the geographical location / time of the year for the comparisons between model and observation. This would also be great to ensure traceability.

* Section 3.2 (In situ observations). I wonder myself whether "Near-surface CO and ozonesondes" could be a more appropriate name, because I assume that most of the aircraft measurements in the previous section will also be in situ.

* Table 4 (summary of bias and correlation coefficients of models vs. aircraft observations) is a bit confusing:

The authors first mention correlation coefficient (R) but then they indicate R-squared both in the caption and the header. Please be consistent.

"Bias and R2a is in pmol/mol, (except for CO and O3)". R or R2 cannot have units. In addition, it might be simpler to write "Bias in pmol/mol (nmol/mol for CO and O3)" and remove the note below the table.

The authors show methyl hydroperoxide (CH3OOH) in the table because there are some aircraft observations, but this species is hardly discussed in the text. Why is that? I think there is only one comment about this species around lines 372-374 ("… while model versions have more difficulties with CH3OOH"), but it is a little bit out of place there. The negative biases can be high for this species but the correlations are fine.

The concept of weighted biases and correlations has been introduced on page 15, but shouldn't the weighted vs. non-weighted values be discussed in some parts of the text? For instance, in the case of C2H4, the three simulations have similar values of R (0.54-0.58) but rather different values for weighted R (0.03-0.39); why is that? In other cases, such as for CO and C2H6, the values of weighted R are higher than those of R. So the weighting can improve or deteriorate the correlations. Summarising, it might be good to explain why the values of R sometimes change so much, not always in the same direction, after weighting.

* Around lines 395-397 the authors say "MOZ simulates too high O3 concentrations ... Here it is worth mentioning that recent updates to reaction probabilities and aerosol radius assumptions in the heterogeneous chemistry module in IFS(MOZART) significantly improved O3 concentrations particularly in the NH".

It might be worth to mention whether the assumptions made in the heterogeneous chemistry are very different in MOZART and CB05BASCOE. Are the uptake coefficients very different in both configurations? Are they constant or parameterized? And which are those aerosol radius assumptions, considering that both configurations use the same CAM aerosol fields?

By the way, the changes mentioned for O3 in the NH, if they are partly consequence of updates in the heterogeneous uptake of N2O5, might have some seasonality and not be so relevant all year round (see e.g. Tie et al., 2003, which used a very simple parameterization of that process). Or do the "recent updates to reaction probabilities" refer to reactions in the gas phase?

Tie, X., L. Emmons, L. Horowitz, G. Brasseur, B. Ridley, E. Atlas, C. Stround, P. Hess, A. Klonecki, S. Madronich, R. Talbot, and J. Dibb: Effect of sulfate aerosol ontropospheric NOx and ozone budgets: Model simulations and TOPSE evidence, J. Geophys. Res., 108(D4), 8364, doi:10.1029/2001JD001508, 2003.

* Figure 4 and Lines 405-406 (comparison with ozonesondes): "However, it should be noted that in the SH regions this evaluation is less representative because there are very few observations".

I see that the number of sites within the different latitude ranges is shown later (Figure 5), but some additional information about the geographical data coverage for this and other datasets  would be useful (e.g. in Section 3, Observational datasets). The authors should also indicate if the ozonesonde dataset is for 2011 or corresponds to the climatology for a longer period. And why not including a map (in a supplement if you prefer) indicating the location of all ozonesonde sites, with a different colour to highlight those shown in Figure 6?

By the way, this sentence in the caption of Figure 4:

"Tropospheric ozone profiles of volume mixing ratios (ppbv) against by model versions CBA (red), MOC (blue) and MOZ (green) against sondes (black) over six different regions" should be rewritten to something like:

"Tropospheric ozone profiles (volume mixing ratios, ppbv) from model versions CBA (red), MOC (blue) and MOZ (green) as well as sondes (black) over six different regions"

* Figure 5. Please draw a line at 0 to better distinguish positive and negative biases. I would also add a symbol to better identify the mean bias within each latitudinal band.

* Lines 454-455 (for modelled O3 vs. MOPITT CO, Figures 7 and 8): "low bias over the NH during April (further analysed in, e.g., Stein et al., 2014), as well as a low bias over Eurasia during August". Need to change "low bias" to "negative bias". In addition, I would say that in August the negative biases occur not only over Eurasia, but over most continental areas of the northern hemisphere.

The authors could also be a bit more quantitative and provide spatial correlations between modelled and observed CO for each of the maps on Figures 7 and 8. They could even show maps with biases in a supplement if some of the biases I mention are not so clear.

By the way, apart from Stein et al. (2014), the authors could also cite this previous paper regarding the low CO bias over the NH:

Shindel et al., Multimodel simulations of carbon monoxide: Comparison with observations and projected near-future changes, 2006, https://doi.org/10.1029/2006JD007100.

* Lines 563-565: This sentence is a bit awkward: "Considering its strong diurnal cycle, due to the fast photolysis rate, here only daytime values have been used to construct the averages, because the observations from the various field campaigns were equally conducted in daylight conditions".

It would be convenient to split all this information into two separate sentences. Moreover, the authors should mention not only the fast photolysis rate but also the strong diurnal cycle of anthropogenic emissions.

* Lines 578–582: "Figures 16 and 17 evaluate tropospheric NO2 using the OMI satellite observations. The simulations deliver generally appropriate distributions … Another interesting finding is a relatively strong negative bias over ...". It is a bit hard to see the biases in the figures. It would be helpful to show analogous maps with those biases (in a supplement).

* Line 600: "Compared to the trace gases previously analysed, Nitric acid is not primary emitted but is purely photochemically formed". Reformulate this sentence, because O3 is not primarily emitted. Also, change "Nitric" to "nitric".

**Technical corrections**

* line 28: Remove comma in "Other, common".

* line 56: "correlation between tropospheric composition with ENSO conditions". Change to

"relationship between tropospheric composition and ENSO conditions" or

"dependence of tropospheric composition on ENSO conditions"?

* line 57: arctic --> Arctic

* lines 65-66. Change "and, afterwards, their chemistry modules technically integrated" to ". Afterwards, their chemistry modules were technically integrated".

* line 85: "which is further expanded" --> "which are further expanded"

* line 130: I think Lacressonnière et al. 2012 is missing in the reference list.

* lines 139-140: "and 2 for the troposphere for the aqueous oxidation reaction of sulfur dioxide into sulfuric acid"-->"and 2 for the aqueous oxidation reaction of sulfur dioxide into sulfuric acid in the troposphere"

* Line 186: Brasseur et al. (1998) is missing in the reference list.

* line 222: "This has consequences for the description of the various emissions". Do you mean "partitioning" instead of "description"?

* Line 246: "experiment ID's" --> "experiment IDs"

* Table 3. Remove "in" from "Specifications of the experiments in evaluated.

* Line 324: No need comma before "over the NH extra-tropics"

* Lines 326, 327 and probably other parts of the text: Please change "highest" to "the highest"

* The expression "associated to" is used three times on page 13 as well as in other parts of the text. I think the use of "associated with" is preferable.

* Line 367: Need space before parenthesis in "deviation(Jöckel".

* line 389: compare --> compares

* "Southern Hemisphere (SH)" is spelled out on line 404, but this should have been done before (around line 393).

* Figure 9: "Izaa" should be changed to "Izaña" (or "Izana" if the "ñ" is problematic) in one of the panels.

* Figure 10: "monthly mean surface CO by observations (GAW network) and by the model simulations" --> "monthly mean surface CO as derived from observations (GAW network) and model simulations"

* line 482: "phae" --> "phase"

* caption of Figures 12, 13, 14, 18 and 19: "Comparison simulated" --> "Comparison of simulated"

* page 27: The reference "Aydin et al., 2014" is mentioned twice. I believe this paper is from 2011, as indicated in the reference list.

* lines 549-550: "the correlation showed acceptable values for all versions (R2 >0.7)"· The authors should explicitly refer to the "weighted correlation" rather than just "correlation".

* lines 562-563: "due to its photochemical balance with nitrogen oxide". Do you refer to NO? If so, please change "nitrogen oxide" to "nitrogen monoxide" or "nitric oxide".

* line 569: Need comma or semicolon before "hence".

* Caption of Figure 16: "Tropospheric NO2 from OMI" --> "Tropospheric NO2 columns from OMI"

* Caption of Figure 17: Remove ", and the model biases with respect to this".

* line 646: Change "is" to "are" in "the averaged differences for O3 (CO) is"

* Reference list: Benedetti et al. (2009) is included twice.

* Reference list: I think that Horowitz et al. (2003) is not mentioned in main text.

* The main text includes these four references: Madronich and Flocke, 1997; Madronich and Flocke, 1999; Madronich et al., 1989; Madronich, 1987. They are somewhat different in the reference list. Please double-check.

---

## Short Comment (SC1) · 30 Jan 2019

Dear authors,

please note, that if only one model is concerned, the title of a GMD manuscript should state the model name (or its acronym) and a version number. These are always important to know even in the case of an evaluation, as different versions might perform differently for the same evaluation procedure. Therefore please change the title of your manuscript accordingly upon revision; e.g., "Quantifying uncertainties due to chemistry modeling – evaluation of tropospheric composition simulations in the CAMS model (version x.y)"

[Figure]

Additionally, please note that evaluation papers also need to include a code availability section, telling the reader how to access the exact code version of the evaluated model or providing profound reasons why the code can not be accessed. Furthermore, please provide reasons, why the data is not freely available.

Best regards,

Astrid Kerkweg
* * *

---

## Referee Comment (RC2) · Anonymous Referee #2 · 28 Feb 2019

This manuscript evaluates tropospheric composition simulations in the CAMS modelling system and quantifies uncertainties related to different chemical schemes. It is well structured and written and illustrates original and interesting results for the . CAMS modelling system. I suggest acceptance of the manuscript for publication after taking into consideration the following comments.

Main Comments 1) I guess that the simulations were carried for the year 2011 but I think the authors should describe in Section 2.3 which was the time period that the simulations were carried out. 2) The authors mention that the averaging of large number of measurements over space and time partly solves the problem of interannual

variability (lines 273-275 in page 11). Can this dataset of Emmons et al. (2000) be representative to compare with the CAMS simulations for the year 2011? I understand the uniqueness of this dataset but could the authors clarify this issue and discuss the uncertainties and the weaknesses of this comparison? 3) I would suggest the authors to provide a short description of the method used to calculate the weighted values of bias and correlation in Table 4. 4) The authors write in line 375 (page 15) that " CBA is the only model version to deliver a satisfactory bias" . Is this a robust conclusion? What is statistically satisfactory? Looking Table 4 I see that in some species CBA bias is smaller than in other schemes, in some other species the biases are comparable and in other species the CBA bias is worse. 5) In the evaluation of ozone the authors conclude that "overall, the evaluation at individual station provides reasonable agreement between model simulations and sondes". How these evaluation results compare with other ozone evaluation studies which were based on MACC and CAMS products (e.g. Inness et al., 2015; Katragkou et al., 2015; Akritidis et al., 2018). I think this conclusion could be also supported by these studies. 6) In lines 448-449 (page 20) the authors write "Approximately half of the CO burden is directly emitted, and the rest formed through degradation of methane and other VOC's". Please add a relevant reference. 7) On how many data points (and years) the temporal correlations shown in Figure 10 are based? 8) In lines 526-528 (page 26) it is written "The vertical profiles (see Figure 13) are strongly biased (e.g., SONEX, Newfoundland and PEM-Tropics-A, Tahiti), with positive biases occurring at the surface and negative in the free troposphere." Could this result also related to inadequate outflow from the atmospheric boundary layer (ABL) to the free troposphere (FT) and hence to model weakness in ABL-FT exchange? 9) The authors refer to correlation R (that span from -1 to 1) but showing $R^2$ which practically describes the explained variance. Although this is not crucial in the discussion it could propagate a misunderstanding on these statistical parameters when the article is published. I would suggest to modify this accordingly. 10) Generally, I think that the discussion in model difference is rather technical and I would suggest the authors to discuss also the possible scientific reasons for discrepancies among the

simulations with different chemical schemes for the different chemical species. Minor comments page 13, line 348: should rather be "relative shorter" instead of "relative short"

---

## Author Comment (AC1) · 1 Apr 2019

**Response to Reviewer comments by Carlos Ordóñez.**

First, we would like to thank the reviewer, Carlos Ordóñez, for his efforts to critically read the manuscript, and provide us with many useful comments and suggestions. Below we answer them to our best ability. The reviewer comments are in italic. Our responses are in regular font, and changes to the manuscript are given in bold.

*The manuscript reports an evaluation of tropospheric reactive gases, including ozone and precursors, in three chemistry versions of the IFS: CB05BASCOE (CBA), MOZART (MOZ) and MOCAGE (MOC). The three configurations use the same meteorology, emissions, transport and deposition schemes. Therefore, the differences in the model results should be basically due to differences in the chemical schemes, namely gas phase chemistry, heterogeneous chemistry, photolysis and chemical solver.*
*Overall, the manuscript is well-written, the presentation quality is good and the authors give credit to related work. As motivated in the introduction, this work documents the diversity in global model simulations caused by chemistry, unlike many other papers which focus on the impact of emissions or other model parameterizations. This is a useful exercise that falls very well within the scope of the journal.*
*My only concern is that I miss some in depth discussions about the potential reasons for the differences in the performance of the three model configurations considering what is different in them (i.e. the chemistry). The "Main comments" below include some suggestions on how to improve this. These are not really major comments. I hope that some of them are useful, but I am aware of the complexity involved in comparing different chemistry schemes. There are also some minor and technical corrections which need attention. I will fully support the publication of the manuscript in GMD once these issues have been addressed.*

*Main comments*
*(1) Section 2.1.4 (Key differences in chemistry modules, including Table 1) provides a relatively detailed overview of the differences in the three chemical mechanisms. All this information is very valuable but not completely helpful. A simple summary comparing the complexity of the different processes in the three configurations could be useful to get a broader picture of the main differences. For instance:*
*1.1. Complexity of organic chemistry: MOCAGE (more extended lumping) ~ MOZART (more explicit) > CB05BASCOE*
*1.2. Complexity of tropospheric inorganic chemistry: MOCAGE (e.g. includes HONO) > MOZART ~ CB05BASCOE*
*1.3. Complexity of tropospheric heterogeneous chemistry (HO2 & N2O5): CB05BASCOE ~ MOZART > MOCAGE.*
*Table 1 indicates that MOCAGE has as many heterogeneous reactions as the other schemes, but that information is not very helpful. I assume those reactions are only relevant for the conversion of SO2 to H2SO4 in the troposphere (lines 139-140) and for stratospheric chemistry, because on lines 179-180 the authors indicate that "Heterogeneous reactions of HO2 and N2O5 on aerosol … not in the IFS(MOCAGE) version considered here".*

We agree with the reviewer that this information is useful. A concern has been how to provide such information in a compact way. We now introduce entries to describe organic and inorganic chemistry, as well as level of complexity regarding aerosol interaction (heterogeneous chemistry and photolysis). The number of heterogeneous reactions is now skipped, as this indeed refers essentially to the stratospheric chemistry, which is not subject of this manuscript.

*1.4. Is there a two-way coupling between MOZART / CB05BASCOE oxidants and CAMS aerosol fields? In other words, are the oxidants passed to the aerosol module to produce aerosols, depleted in that processes and then passed back to the chemistry schemes? Or does the coupling only consist of using CAMS aerosols for heterogeneous chemistry in the two chemistry modules? This information is not clear in the main text and is relevant e.g. to evaluate differences in NOx and HNO3.*

None of the current versions contain explicit parameterizations for two-way coupling with the CAMS aerosol fields; this is actually covered in more recent versions of the modeling system. Aerosol fields are currently only used for defining the surface area density needed in the heterogeneous chemistry. We now write

**Also two-way coupling of secondary aerosol formation was not available in the current model version.**

*1.5. Complexity of photolysis: Is it more complex in CB05BASCOE because it includes the impact of aerosols? Anyway, if there is no easy way to compare complexity of photolysis among the three configurations one can leave this out.*

We now specify the aerosol impact on photolysis. Other aspects are more difficult to specify in the table.

*1.6 Complexity of stratospheric chemistry: CB05BASCOE > MOZART ~ MOCAGE*

We choose to focus on tropospheric chemistry aspects mainly in the table, so we prefer to leave out a comment on this.

*1.7 Chemical solver: CB05BASCOE uses Rosenbrock (differences in NOy chemistry and up to ~20% less O3 regionally). It would be interesting to know if the "differences in the N2O5 chemistry" indicated by the authors (line 199) often result in smaller amounts of the species evaluated here (NO2 and HNO3). If not clear, please leave this out.*

This is correct: The introduction of the more accurate Rosenbrock solver was found to lead to a reduction in NO2 and HNO3, associated to a faster N2O5 production, together with an equally stronger N2O5 heterogeneous loss on aerosol particles. However, as this process is indeed connected to application of heterogeneous chemistry, this effect is only relevant to CB05BASCOE (and MOZART), and not to MOCAGE chemistry. We now describe these processes more explicitly:

**These differences are mostly traced to an increase in the $N_2O_5$ chemical production (Cariolle et al., 2017), reducing in turn the $NO_x$ lifetime because of a larger net $N_2O_5$ loss on aerosol. This in turn leads to a reduced chemical ozone production efficiency.**

*Summarising, although I understand how hard it is to compare different chemistry schemes, a new table or some bullet points similar to those indicated above could be used to better see the main differences in the three model configurations. Afterwards, one could elaborate on the potential implications for the representation of the main tracers evaluated here (see next comment).*

We expanded the table and its description in response of the reviewer comments as described above.

*(2) Some additional comments in different parts of the text could be helpful to try to understand the model differences. As mentioned above, I do not expect the authors to be able to explain absolutely everything, but hope that some of these comments will be useful and easy to include in the text:*

*2.1 Section 4 (Assessment of inter-model differences). The comparison of NOx annual mixing ratios (Figure 2, lines 330-333) could easily be improved:*

*"MOZ and CBA are overall similar, but MOC is showing higher values over the NH high-latitudes (>60°N) and also at altitudes below 900 hPa in the tropics". First, a minor correction: I clearly see differences in the lower to mid-troposphere from the tropics to the high latitudes, i.e. not only for latitudes >60°N and below 900 hPa in the tropics.*

We agree with the reviewer that this is actually better, we modify the text accordingly.

*"This is likely related to the fact that in this version of IFS(MOCAGE) the coupling with the aerosol module has not yet been established, contrary to CBA and MOZ". Do you mean that the lack of heterogeneous hydrolysis of N2O5 in MOC will probably raise the NOx concentrations (because that reaction produces the reservoir species HNO3, which is efficiently removed from the atmosphere by dry and wet deposition, resulting in a removal of NOx)? If so, please clearly indicate this. Otherwise, you should explain which other processes might be involved. For instance, is there aerosol nitrate production, removing NOy from the gas phase, in CBA and MOZ?*

Yes, here we indeed exactly refer to the differences in heterogeneous reaction approaches, as we now more explicitly describe in the table. We now write this explicitly:

"This is likely related to the fact that in this version of IFS(MOCAGE) the coupling with the aerosol module has not yet been established, contrary to CBA and MOZ**, implying a missing sink of NOx through the heterogeneous reaction of $N_2O_5$ to $HNO_3$**."

*2.2 The last paragraph of Section 4 (Assessment of inter-model differences) tries to relate the differences in NOx-O3-OH in the models. Overall, the authors point to faster chemical production of O3 in MOCAGE, counterbalanced by fast loss by reaction with OH and HO2. They also suggest a relatively short lifetime of O3 in MOCAGE and conclude the paragraph with "Such differences in oxidation capacity naturally have important implications for understanding differences in the performance of NMHCs, as discussed in the next sections". A couple of questions:*

*\* The authors are probably right, but does the C-IFS have any diagnostics that can be used to quantitatively compare budgets/lifetimes of O3 due to different processes in the three configurations? That would be useful.*

Unfortunately such diagnostics, keeping track of chemical ozone and carbon monoxide production and loss budgets, are only available for CB05BASCOE chemistry, and not for the other chemistry schemes, hence we cannot easily intercompare them here. It is a good suggestion to actually implement such diagnostics for the other chemistry versions as well. We now include the following comment:

**An assessment of the ozone chemical production and loss terms is beyond the scope of this work.**

*\* That paragraph should also include a comment for CO. A few lines above (lines 326-328), the authors compare CO in the three configurations (CBA >> MOZ > MOC), "suggesting mainly differences in oxidizing capacity". I think they could be more specific. I am aware that there is no straightforward comparison of the CO and OH levels in the models, because different VOC degradation schemes will impact the CO mixing ratios (as correctly indicated on first paragraph of Section 5.2). Nevertheless, the authors could relate the higher (lower) CO levels in CBA (MOC) to the lower (higher) OH levels seen for the same models in Figure 2. I will come back to this below (comment on Section 5.2).*

The reviewer is correct that a large portion of the spread in CO is likely caused by differences in OH, considering that CO and precursor emissions are mostly identical. Unfortunately we do not track the CO chemical production and loss budgets, so we cannot quantify this easily. Still, in the section describing CO we now point explicitly to the impact of OH, and refer to its discussion:

**As CO and precursor emissions are essentially identical, this is likely caused by differences in oxidizing capacity which is governed by OH abundance, as described below.**

*2.3. Section 5.1 provides a detailed evaluation of ozone. However, after so many comparisons, it is a bit hard to conclude what we have learnt. A summary of the model performance, even if the reasons for the model differences need to be better investigated, might help.*
*For instance, I know there are some specific comments for MOZ in the first paragraph of the section (e.g. "MOZ simulates too high O3 concentrations ... Here it is worth mentioning that recent updates to reaction probabilities and aerosol radius assumptions in the heterogeneous chemistry module in IFS(MOZART) significantly improved O3 concentrations particularly in the NH"), but a summary mentioning that at the end would be useful.*

*Some conclusions might be that modelled ozone is often within the uncertainty of the selected sondes (Figure 6), but that MOZ tends to yield higher O3 levels everywhere in the mid-to-upper troposphere (500-300 hPa) and everywhere in the northern hemisphere (Figure 5). Somewhere one should also mention that this configuration yields the highest biases (15.9 ppb), but at the same time the best weighted correlations, when compared to aircraft data (Table 4). Then, one could say again that the biases have partly been corrected in recent versions with improved heterogeneous chemistry.*
*Finally, the authors could try to repeat Figure 5 but showing accumulated errors (e.g. RMSE) instead of biases. If that provides any further information to document the performance of the three configurations then it should be included.*

We now include figures on the RMSE within Figure 5, and link this to a summary section to the discussion of the ozone evaluation. For this purpose we have slightly re-ordered the section, to describe the findings from Figure 5 (new Figure 7) at the end of the section. Also, as per the second reviewer comments, we include an evaluation against the CAMS Interim Reanalysis, to put this evaluation into further perspective. We now write:

**Evaluation against the aircraft climatology as provided in Table 4 shows on average a positive bias in the range 10 (CBA and MOC) to 16 (MOZ) ppbv, while the correlation statistics shows generally acceptable values ($R^2 > 0.57$), giving overall confidence in the model ability to describe ozone variability. Figure 6 shows annually averaged model biases and root mean square errors (RMSE) for various latitude bands and for altitude ranges 900-700hPa, 700-500hPa and 500-300hPa against WOUDC sondes. In this**

**evaluation we also present data from the CAMS Interim Reanalysis (CAMSiRA) for the year 2011, to put the current model evaluation into perspective. This summary analysis shows averaged biases within ±10 ppbv, which is also in line with the O3 bias statistics against the aircraft climatology. At lower altitudes the model biases are mostly equal or better than those from CAMSiRA, while above 500 hPa CAMSiRA delivers mostly smaller biases thanks to the assimilation of satellite ozone observations. The RMSE shows a larger spread in the lower troposphere of the NH, while at higher altitudes, above 500 hPa the overall magnitude of the RMSE for the three chemistry versions converges to values ranging from 10 to 16 ppbv, depending on the latitude. Here the CAMSiRA shows overall better performance, mainly for the tropics and SH, while over the NH its performance is similar to IFS(CBA). This evaluation summarizes common discrepancies between model versions and observations, such as the negative bias over the Antarctic and positive bias below 700 hPa for tropical stations (see also Figure 4), suggesting biases in common parameterizations such as transport, emissions and deposition. The largest discrepancies between model versions have been detected at northern mid- and high latitudes below 500 hPa, with significantly higher values for RMSE for MOC and MOZ compared to CBA. A comparatively large positive bias for MOZ was detected, which has been linked to an under-estimate of the N2O5 heterogeneous loss efficiency. The differences between MOC and CBA can likely be explained by similar aspects are likely as important to explain differences with respect to the performance of IFS(MOCAGE).**

*2.4. Section 5.2 (Carbon monoxide).*
*The authors include a useful introduction about all the processes impacting CO levels (emissions and VOC degradation) and then comment several plots comparing models with different datasets (MOPITT, surface, aircraft profiles). Overall, I am rather happy with this section, but again it would be nice to finish it with a paragraph summarising what we have learnt from these comparisons.*
*For instance, from Figures 7-9 (MOPITT & surface CO) one can see that CBA is the configuration with the highest CO levels, yielding the best comparisons (lowest negative biases) in the northern hemisphere but positive biases in the southern hemisphere (Figure 9). The authors should mention that this may partly be related to the fact that this scheme is the one with the lowest OH levels (Figure 2). I would say this is one of the main outcomes from these comparisons.*
*Towards the end of this section the authors could indicate whether modelled CO is within a given range for the three schemes (they mention 10% only for the comparisons with aircraft observations, but in the abstract and conclusions they indicate 20%). That range (expressed as a percentage or ppb) could be compared to the overall model error for this species (when compared e.g. with surface observations) to get an idea about the fraction of that error that could be attributed to differences in the chemistry. Here or in the "Conclusions" section one could even point to the need of producing some model diagnostics in the future to understand to what extent the differences in the atmospheric mixing ratios of CO arise from differences in CO degradation (by reaction with OH) or differences in CO production (from VOC degradation) in the chemistry schemes.*

In response to the reviewer we now make the relevance of OH for explaining the differences in CO clearer. The discrepancy in reporting of the 10% to 20% spread in CO at different parts in the manuscript is because of the different means to quantify this. The 20% number originates from assessment of the zonal mean CO fields (Sect. 4), while the 10% specifically relates to the model CO profiles presented in the evaluation against specific aircraft observations (Sect. 5.2). Closer assessment of the evaluation of vertical profiles shows that this is rather in the range 10-20%, depending on the location. We now modify the text accordingly.

*2.5. Section 5.3 (formaldehyde). It is nice that the authors use weighted biases to show that the simulations are within the range of uncertainty of the few observations available. The correlations are also quite decent. However, this section is rather short and could probably benefit from some improvements and/or discussions.*

*Lines 506-508 say: "Formaldehyde is important as the most ubiquitous carbonyl compounds in the atmosphere (Fortems-Cheiney et al., 2012). It is mainly formed through the oxidation of methane, isoprene and other VOC's such as methanol (Jacob et al., 2005), while its oxidation and photolysis is responsible for about half of the source of CO in the atmosphere".*

*First, a couple of minor comments: You should mention that HCHO is not only secondarily formed but also directly emitted. In the first sentence, do you mean "one the most ubiquitous carbonyl compounds"?*

*I am not sure that the end of the second sentence is right. There the authors mention that the degradation of HCHO is half the source of CO in the atmosphere. Somewhere else (lines 448-449) they mention that "Approximately half of the CO burden is directly emitted, and the rest formed through degradation of methane and other VOCs". Are the authors implying that most of the CO formed (not emitted) in the atmosphere comes from HCHO? Isn't that too much? Before making that statement, the authors should check some references or compare the production terms of CO in the chemistry schemes. If HCHO is so important in terms of CO production, could one try to relate the model performance for both species? For instance, it is possible to relate the fact that CBA is the simulation with the highest CO mixing ratios (which I previously attributed to the low OH in that model) to the negative HCHO biases found for the same scheme in this section?*

In response to the reviewer, first of all, we should indeed have written "**one of the most ubiquitous carbonyl compounds**"; we do this now.

The reviewer is in principle correct that $CH_2O$ is also emitted, with an amount of ~13 Tg yr$^{-1}$, see Table 2 in the manuscript. However, these direct emissions are negligible compared to the secondary production through VOC oxidation: from the chemistry budget analysis as available in the CB05BASCOE scheme this amount is estimated to be ~ 1550 Tg yr$^{-1}$, with contributions indeed from methane, isoprene, methanol, and many other VOC's. $CH_2O$ degradation through photolysis and oxidation in turn corresponds to approx. 1300 Tg CO production. The remainder of secondary CO production from other VOC oxidation is estimated about 250 Tg, while direct CO emissions are 1040 Tg, which explains our statement that $CH_2O$ degradation is responsible for about half the source of CO.

However, the amount of modeled chemical $CH_2O$ production is indeed depending on the chemistry mechanism. Note additionally that the OH impacts both on $CH_2O$ production, and its loss. Therefore, evaluation of the $CH_2O$ in order to understand CO biases, is not trivial. As any explanation is essentially speculation we refrain from further discussion. Indeed, this requires information on the chemical production and loss budget terms, which should be made available for all chemistry versions. The same argument holds for $CH_3OOH$, as further discussed in a reviewer comment below. We now include the following comment:

**Considering the short lifetimes for $CH_2O$ (a few hours in daytime) and also $CH_3OOH$, and the large dependence of their abundances on details of the VOC degradation scheme which vary across the chemistry versions presented here, it is beyond the scope of this manuscript to explain these differences. This would require a detailed assessment of the respective production and loss budgets which are currently not available.**

*2.6. Section 5.4 (Ethene, C2H4).*
*The authors have mentioned the main sources of this species and have provided some insights into the reasons for the model performance. Most of this section looks good to me, but I have some comments:*
*\* The last paragraph says: "Furthermore, interesting is the comparatively large difference present between the simulations at high latitudes (e.g. SONEX, Newfoundland), where the largest relative differences in modelled OH have been found, (see also Sec. 4), illustrating the importance of OH for explaining inter-model differences". Could one explicitly mention that the higher C2H4 mixing ratios in CBA (red in Figure 13, right) compared to the other two schemes are probably related to the lower OH levels in the same scheme?*

We now explicitly mention this correlation between the relatively high C2H4 in relation to the low OH:

**CBA indeed shows the largest values for $C_2H_4$, which is correlated to the comparatively low magnitude of OH, which explains the longer lifetime for $C_2H_4$.**

*On the other hand, I am surprised that CBA is the scheme with the largest negative biases for C2H4 in Table 4 (-6.28 ppt). It does not look like that from Figure 13. Is that because the figure only shows data for some specific campaigns and locations? If so, my previous comment might not be valid.*

In fact the reviewer has spotted a typo when generating Table 4. The reported overall model bias should not have been negative but positive by 6.28 ppt for CBA. This is corrected now, which makes the evaluation more consistent. Indeed, the reviewer comment still holds that here we show figures for a few locations; for other sites the magnitude of the biases vary to some extent.

*\* Lines 531-532: "As the underestimation appears to be ubiquitously distributed this suggests that C2H4 decomposition is too strong, or that the model versions miss chemical production term (e.g., Sander et al., 2018)". This conclusion looks fine, but they should change "miss chemical production term" to "miss some chemical production terms".*

The reviewer is correct, we change accordingly.

*Somewhere in this section it would be worthwhile to briefly mention (without all the explanations given below) that one should not expect to get this species right because model and observations are not really collocated in time. I understand that model data are monthly averages (considering both daytime and night-time data) while observations were probably collected at daylight. In addition, model and aircraft observations are from different years. All this is relevant if we keep in mind all the processes affecting the mixing ratios of this species.*

The reviewer is correct that no exact correspondence should be expected for the evaluation against the aircraft observations, particularly considering the different model year. This has been discussed in Sec. 3.1. To better explain the procedure for the model evaluation, we now add in Sec. 3.1 the following comment:

Aircraft measurements of trace gas composition from a database produced by Emmons et al. (2000) were used for evaluation **of distributions of collocated monthly mean modelled fields**.

While in Sec. 5.4 we add the comment:

**Even though this evaluation should only be considered in a climatological sense, …**

*Many of those processes have already been mentioned by the authors: terrestrial biogenic emissions (probably with strong diurnal cycle), which should be much larger in Newfoundland (Figure 13 right, high C2H4 mixing ratios) than in Christmas Islands and Tahiti (oceanic environment, Figure 13 left and centre, low C2H4); diurnal cycle of anthropogenic emissions (again source probably much higher in Newfoundland); biomass burning sources (with strong interannual variability); upward lifting and subsequent horizontal transport of plumes (not sure if seen on observations from Figure 13 left for Christmas Islands), and relatively short lifetime, mainly controlled by its reaction with OH (higher levels at daylight than at night-time).*

The reviewer is correct, but for reasons of brevity we prefer not to include such a listing here.

*2.7 Section 5.6 (Nitrogen dioxide, NO2).*

*Overall, this section is fine too, but I miss more detailed explanations and some integration with the rest of the text towards the end of the section. In addition, providing maps with biases would be helpful to see the differences between the model configurations and OMI (this is also indicated in a minor comment).*

We now include bias plots instead of actual model fields for tropospheric NO2 columns, to better assess the differences with respect to the retrievals.

*For instance, around lines 582-586, the authors mention the lower NO2 levels for CBA than for the other two models, over polluted and clean regions in the northern hemisphere. Then, around the same lines they say: "This all suggests a relatively short NOx lifetime in CBA compared to MOZ and MOC, which in turn helps to explain the lower O3 over the NH-mid latitude regions as modelled with CBA (see Figure 5)". A couple of comments on this:*
*\* In Section 2.1.4 the authors mentioned some implications of the use of the Rosenbrock solver (in CBA) for NOx and O3, but I think that is hardly mentioned again in the rest of the text. Shouldn't one mention it here?*

The reviewer is correct that the use of different solvers is an important aspect to explain the differences, particularly for IFS(MOCAGE) but it is not the only explanation. Also the $N_2O_5$ heterogeneous chemistry, missing in this version of IFS(MOCAGE) and not optimized in IFS(MOZART) contributes to various degree. To point this out, we now include the following comment:

**The causes of these differences in modelled $NO_2$ are mainly the use of a different numerical solver and differences in the efficiency assumed for $N_2O_5$ heterogeneous reactions (see Sec. 2.1.4).**

*\* CBA has less NOx, O3 and OH, but more CO, than the other model configurations. Not sure if all these facts are related. Shouldn't this be mentioned somewhere?*

These aspects are certainly related. This has been discussed in Sec. 4, describing an overview of the main differences. We prefer not to describe such correlations here again – the subsections in Sec. 5 are mostly focused on the quality of individual species.

*2.8. HNO3 (also related to NOx).*
*There is a good introduction relevant to HNO3 around lines 601-604: "It has a very high solubility and therefore tends to be scavenged by precipitation very efficiently, providing an effective sink for the NOx family. Furthermore, it can act as a precursor for nitrate aerosols (Bian et al., 2017). HNO3 concentrations are therefore expected to show amongst the largest variation between the simulations, as the production and the sink terms can largely differ due to uncertainties in the parameterizations". However, later on, the authors are not very specific about the potential reasons for the model differences they find. They just add some general sentences like these ones:*
*Lines 612-613 (for HNO3): "The discrepancies between the model versions can be mainly attributed to differences in NOx lifetimes and nitrate aerosol formation"*
*Lines 653-655 (in the conclusions): "For instance IFS(MOCAGE) tends to predict significantly higher NOx and HNO3 concentrations in the lower troposphere compared to the other two chemistry versions".*
*Can one relate some of the model differences to the lack of coupling of MOCAGE chemistry with the aerosol module? I assume this is not simple. For instance, the inclusion of heterogeneous loss of N2O5 in CBA and MOZ should remove NOx from the system but form HNO3, which would not explain the high HNO3 in MOC. On the other hand, if there is a two-way coupling between the chemistry and the aerosol modules, CBA and MOZ will provide oxidants (e.g. HNO3) to the aerosol model to form aerosols (e.g. nitrate) and the oxidants will be depleted in that process. This could counteract the production of HNO3 by heterogeneous reactions. Are all these processes taking place in CBA and MOZ but not in MOC?*
*And is wet deposition done exactly in the same way for the three configurations? Otherwise, it should have an impact on the HNO3 mixing ratios.*

In response to the reviewer, we confirm that interpretation of differences in $HNO_3$ is not straight forward, exactly because it is a secondary product. We note that dry and wet deposition parameterizations, and loss rates, are identical across the three versions.
Aspects that may play a role are in leading to differences are, once again, a larger $HNO_3$ production efficiency in CBA and MOZ, compared to MOC, which currently misses the $N_2O_5$ heterogeneous loss reaction. As this process mostly takes place near the surface, any $HNO_3$ production here consequently may lead to a more efficient dry (and also wet) deposition of $HNO_3$, as compared to $HNO_3$ produced by $NO_2$ oxidation at more remote locations in the free troposphere.
Also the versions include different parameterizations for secondary aerosol formation (nitrate and ammonium), for which we now provide details in Sec. 2.1.1 for IFS(CB05BASCOE) and 2.1.3 IFS(MOZART). Note that the respective chemistry versions contain separate tracers to describe the nitrate aerosol; they are not yet part of the CAMS aerosol module.
However, as it is speculation as to what are the relative contributions of the various mechanisms to the observed model spread, we limit ourselves by adding the following statement to this section:

"The discrepancies between the model versions can be mainly attributed to differences in NOx lifetimes, **associated to differences in heterogeneous chemistry, and parameterizations for nitrate aerosol formation, as discussed in Sec. 2.1.4**."

While in Sec 2.1.4 we now describe in more detail the differences in nitrate aerosol formation modeling and heterogeneous chemistry:

**"Instead, IFS(CB05BASCOE) and IFS(MOZART) contain a treatment of gas-aerosol partitioning for nitrate and ammonium, which is missing in IFS(MOCAGE).**

**Significant uncertainty remains in the magnitude of heterogeneous reaction probabilities.** Heterogeneous reactions of $HO_2$ and $N_2O_5$ on aerosol are included in IFS(CB05BASCOE) and IFS(MOZART), **although with different efficiencies,** but not in the IFS(MOCAGE) version considered here. This has only become available in a more recent model version. **Also two-way coupling of secondary aerosol formation was not available in any of the current model versions.**

*Minor comments*
*\* line 22: "compared against a range of aircraft field campaigns, ozone sondes and satellite observations". Surface observations should also be mentioned here (even if only used for CO).*

Done

*\* Lines 25-26: "Most of the divergence in the magnitude of NMHCs can be explained by differences in OH concentrations". Why only NMHCs? Shouldn't one mention CO too?*

The reviewer is correct: CO should be mentioned here as well.

*\* Chemical names, formulae and abbreviations are not used in a consistent way throughout the text. There are some examples here, but this list is not exhaustive at all (the authors should revise the whole text):*
*"PAN" should be changed to "peroxyacetyl nitrate (PAN)" on line 135.*

Done

*"polar stratospheric clouds (PSC)" needs to be mentioned only once (for B05BASCOE). In the description of the other two configurations the authors can simply use "PSC".*

Done

*It should be "VOCs" and not "VOC's". The same applies to "NMHC's" and "CFC's". Please, also spell out VOC and CFC.*
*In particular, special attention is required in Section 2.2 (Emissions). There the authors use capitals for names such as "Olefins", "Aldehydes", "Butenes" or "Parafins", but that is not needed.*

The reason for using capitals was as these refer to actual tracers in the CB05 mechanism. Instead, we now put brackets around these tracer names.

*The names "methane" and "CH4" are used in different parts of the text, but the authors never indicate "methane (CH4)". All names of chemical species should be given together with their formulae the first time they appear in the text.*

Done

*They authors use the name "acetylene", while the IUPAC name (ethyne, C2H2) might be more appropriate.*

Done

*When possible they should indicate both the name and the formula for all the species in Table 2 (otherwise, please give both name and formula in the main text). There should be a reference to this table from the very beginning of section 2.2.*

Done

*\* Section 2.2. Please indicate whether the non-anthropogenic emissions (e.g. biogenic VOCs and NO, NO from lightning) are prescribed or interactive.*

All emissions, including biogenic and soil NOx, are prescribed. The only exception is lightning NOx emission, which is parameterized depending on convection. We now make this more explicit.

*\* The authors have indicated the experiment IDs in Table 3. This and other information they provide will be very useful to ensure traceability. I have also noted that in the "Data Availability" section they say: "The model simulation data as used in this work can be obtained upon request from the corresponding author". Are the model experiments from Table 3 stored on ECMWF's servers and accessible to other users? If that is the case, the data availability section could be rewritten.*

Data can be retrieved by other users who have access to ECWMF MARS archiving system, indeed using the experiment ID's. Users are welcome to download the data from there. However, as it is not fully straight forward to identify the actual composition fields, it is recommended that potential users contact one of the authors and/or the corresponding author, who can provide further support. We provide such details in the Data Availability section now.

*\* In Section 2.3 (Model configuration and meteorology) the authors mention that "the model was sampled … every three hours. These are used to compute monthly to yearly averages. Standard deviations are computed to represent the model variability for a specified range in time and space". This is fine but, after reading section 3 (Observational datasets), it is not clear how the model output and observations are sampled for many of the comparisons shown in the text. I assume that the data are always collocated in space but not completely collocated in time. For instance, aircraft observations are from different years. Please indicate for all datasets whether they are for the simulated year (i.e. 2011) or for another period. Then indicate whether all comparisons are made using model and observations from the same month (even if the data are from different years).*

We now specify more explicitly the sampling approach for the various observations. For evaluation against the aircraft measurements data for 1990-2001 were used. For this

comparison the model results were co-located spatially and temporally with the observations, although for a different year. Monthly mean surface CO observations, as provided by the World Data Center for Greenhouse Gases (WDCGG) for the year 2011 were used while for the other evaluations (ozone sondes, CO and $NO_2$ satellite retrievals) we use a different collocation procedure as already described in the manuscript. We now write:

Aircraft measurements of trace gas composition from a database produced by Emmons et al. (2000) were used for evaluation **of distributions of collocated monthly mean modelled fields**.
(…) The database is formed by data from a number of aircraft campaigns **that took place during 1990-2001,** …

In-situ observations for monthly mean CO for the year 2011 are used **to evaluate monthly mean modelled surface CO fields**. **Observational data is taken from the World Data Centre for Greenhouse Gases (WDCGG), the data repository and archive for greenhouse and related gases** of the World Meteorological Organisation's (WMO) Global Atmosphere Watch (GAW) programme.
(…)

**The 3-hourly output of the three model versions has been collocated to match to the location and launching time of the individual sonde observations during 2011.**

*\* Section 3.1 (Aircraft measurements). The authors used the compilation of Emmons et al. (2000) as well as TOPSE and TRACE-P data. If it is not too complicated, they could also indicate the names of all the campaigns used, the period covered and so on. In Figure 1 they indicate that "Each field campaign is represented by a different colour" but they do not mention the names. Later, some of those names appear on the profile plots (Figure 11, 12, …). I simply think it would be helpful to know the geographical location / time of the year for the comparisons between model and observation. This would also be great to ensure traceability.*

This is a good suggestion. In Figure 1 we now provide the names of the campaigns in the map presenting the location of the aircraft data, which helps to identify the location of the evaluations.

*\* Section 3.2 (In situ observations). I wonder myself whether "Near-surface CO and ozonesondes" could be a more appropriate name, because I assume that most of the aircraft measurements in the previous section will also be in situ.*

The reviewer is actually correct. We change this now.

*\* Table 4 (summary of bias and correlation coefficients of models vs. aircraft observations) is a bit confusing:*
*The authors first mention correlation coefficient (R) but then they indicate R-squared both in the caption and the header. Please be consistent.*

We change this consistently to $R^2$ in the header, see also the second reviewer comment

*"Bias and R2a is in pmol/mol, (except for CO and O3)". R or R2 cannot have units. In addition, it might be simpler to write "Bias in pmol/mol (nmol/mol for CO and O3)" and remove the note below the table.*

We follow the suggestions from the reviewer, and now write in the table header:

**Bias[a] is given in pmol/mol, (nmol/mol for CO and $O_3$), Bias[b] is in standard deviation units. Likewise, $R^{2a}$ is the normal correlation coefficient, and $R^{2b}$ the correlation coefficient weighted with standard deviations (see text).**

*The authors show methyl hydroperoxide (CH3OOH) in the table because there are some aircraft observations, but this species is hardly discussed in the text. Why is that? I think there is only one comment about this species around lines 372-374 ("… while model versions have more difficulties with CH3OOH"), but it is a little bit out of place there. The negative biases can be high for this species but the correlations are fine.*

We now add a small section describing the CH3OOH evaluation within the section describing CH2O. In short, the reviewer is correct that the model versions are in reasonable shape to represent the variability in CH3OOH, reflected by good correlations. Only IFS(MOZART) is somewhat on the low side with a comparatively large negative bias. Unfortunately as with CH2O, it is very difficult to provide additional interpretation to these model differences, considering that this species, as CH2O, is relatively short lived (governed by OH and photolysis mainly), while its production rate much depending on details of the VOC degradation scheme:

**$CH_3OOH$ is a main organic peroxide acting as a temporary reservoir of oxidizing radicals, Zhang et al. (2012). It is mainly formed through reaction of $CH_3O_2$ + $HO_2$, which are both produced in the oxidation process of many hydrocarbons. The $CH_3OOH$ lifetime of globally about one day is mainly governed by its reaction with OH, and photolysis. Figure 13 presents an evaluation for $CH_3OOH$ for the same sites are presented for $CH_2O$ in Figure 12. Mixing ratios are generally reasonably within the range of the observations, as for example over the tropical Pacific over Fiji. A larger spread between model versions, with a strong over-estimate for CBA, is found in the Amazon region over Brazil. As a global average, a comparatively large under-estimate for MOZ and, to a lesser extent also for CBA, was found, see also Table 4. Nevertheless, correlations, especially those weighted with the uncertainties, are overall good, giving general confidence in the modeling.**

**Considering the short lifetimes for $CH_2O$ (a few hours in daytime) and also $CH_3OOH$, and the large dependence of their abundances on details of the VOC degradation scheme which vary across the chemistry versions presented here, it is beyond the scope of this manuscript to explain these differences. This would require a detailed assessment of the respective production and loss budgets which are currently not available.**

*The concept of weighted biases and correlations has been introduced on page 15, but shouldn't the weighted vs. non-weighted values be discussed in some parts of the text? For instance, in the case of C2H4, the three simulations have similar values of R (0.54-0.58) but rather different values for weighted R (0.03-0.39); why is that? In other cases, such as for CO and C2H6, the values of weighted R are higher than those of R. So the weighting can improve or deteriorate the correlations. Summarising, it might be good to explain why the values of R sometimes change so much, not always in the same direction, after weighting.*

Generally higher correlations can be expected when accounting for the uncertainties, quantified by the standard deviation in the model and observations. In fact, for cases of high variability in either the model or observations, a poor matching between the observations and model can be expected, which causes poor correlation. In the weighted correlation, as these situations are weighted less, an increase in the value for $R^2$ is expected. Part of this argumentation is already discussed when introducing this statistics, in Sec 5.

An exception is the $C_2H_4$, which does not perform well across the chemistry versions with rather different profile shapes compared to the observations. In this case, the weighting with variability apparently does not help, and in fact only leads to further degradation. This indicates more fundamental problems with describing the spatial and temporal variability for species properly, as further discussed in Sec. 5.4.

We now expand on the discussion of normal and weighted $R^2$ and biases by writing in Sec. 5:

**For this reason the weighted correlations are also generally expected to be higher than the normal correlations.**

And:

**Remarkably, $C_2H_4$ is the only trace gas where values for the weighted $R^2$ are lower than the normal $R^2$ values, suggesting fundamental problems representing this trace gas properly in any of the chemistry versions.**

*\* Around lines 395-397 the authors say "MOZ simulates too high O3 concentrations ... Here it is worth mentioning that recent updates to reaction probabilities and aerosol radius assumptions in the heterogeneous chemistry module in IFS(MOZART) significantly improved O3 concentrations particularly in the NH".*
*It might be worth to mention whether the assumptions made in the heterogeneous chemistry are very different in MOZART and CB05BASCOE. Are the uptake coefficients very different in both configurations? Are they constant or parameterized? And which are those aerosol radius assumptions, considering that both configurations use the same CAM aerosol fields?*
*By the way, the changes mentioned for O3 in the NH, if they are partly consequence of updates in the heterogeneous uptake of N2O5, might have some seasonality and not be so relevant all year round (see e.g. Tie et al., 2003, which used a very simple parameterization of that process). Or do the "recent updates to reaction probabilities" refer to reactions in the gas phase?*
*Tie, X., L. Emmons, L. Horowitz, G. Brasseur, B. Ridley, E. Atlas, C. Stround, P. Hess, A. Klonecki, S. Madronich, R. Talbot, and J. Dibb: Effect of sulfate aerosol ontropospheric NOx and ozone budgets: Model simulations and TOPSE evidence, J. Geophys. Res., 108(D4), 8364, doi:10.1029/2001JD001508, 2003.*

The rate of $N_2O_5$ uptake on aerosols, which depends on aerosol composition and meteorological conditions, remains highly uncertain. In order to investigate the impact of such uncertainties on simulated ambient $N_2O_5$ and hence on $O_3$ in MOZ, sensitivity simulations have been recently conducted using reaction probability and aerosol parameters for $SO_4$, OC and BC from MOZART-4 (Emmons et al., 2010). The aerosol parameters and reaction probability applied show significant differences as compared to those in CB05BASCOE (e.g. $\gamma$ values of 0.02 and 0.1 for uptake on $SO_4$ in CB05BASCOE and MOZART-4, respectively). Both MOZ and

CB05BASCOE use constant values for reaction probabilities and mean particle radius while hygroscopic growth rate factors at different RH were applied according to Chin et al. (2002).

The sensitivity simulations showed significant decreases in $N_2O_5$ concentrations due to heterogeneous reactions on aerosols and important decreases in $NO_x$ which led to lower $O_3$ in MOZ particularly in the NH. The changes in $O_3$ and the overall $NO_x$ budget in the NH are found to be more important in winter and spring time periods, consistent with results of Tie et al. (2003). We now include the following comment in the manuscript, sec. 2.1.4:

**Significant uncertainty remains in the magnitude of heterogeneous reaction probabilities.** Heterogeneous reactions of $HO_2$ and $N_2O_5$ on aerosol are included in IFS(CB05BASCOE) and IFS(MOZART), **although with different efficiencies,** but not in the IFS(MOCAGE) version considered here. This has only become available in a more recent model version. **Also, for instance, a more recent version of IFS(MOZ) with updated values following Emmons et al. (2010) leads to a significantly reduced $NO_x$ lifetime.**

*\* Figure 4 and Lines 405-406 (comparison with ozonesondes): "However, it should be noted that in the SH regions this evaluation is less representative because there are very few observations".*
*I see that the number of sites within the different latitude ranges is shown later (Figure 5), but some additional information about the geographical data coverage for this and other datasets would be useful (e.g. in Section 3, Observational datasets). The authors should also indicate if the ozonesonde dataset is for 2011 or corresponds to the climatology for a longer period. And why not including a map (in a supplement if you prefer) indicating the location of all ozonesonde sites, with a different colour to highlight those shown in Figure 6?*
*By the way, this sentence in the caption of Figure 4:*
*"Tropospheric ozone profiles of volume mixing ratios (ppbv) against by model versions CBA (red), MOC (blue) and MOZ (green) against sondes (black) over six different regions" should be rewritten to something like:*
*"Tropospheric ozone profiles (volume mixing ratios, ppbv) from model versions CBA (red), MOC (blue) and MOZ (green) as well as sondes (black) over six different regions"*

Ozone data used here is indeed for the year 2011, as we now specify in Sec 3.2. We now also include a figure presenting the location of ozone sondes used in the evaluation.
Also we have adapted the legend in Figures 4 and 5 according to the reviewer suggestion.

*\* Figure 5. Please draw a line at 0 to better distinguish positive and negative biases. I would also add a symbol to better identify the mean bias within each latitudinal band.*

We now draw a line at zero, to better identify the positive/negative biases.

*\* Lines 454-455 (for modelled O3 vs. MOPITT CO, Figures 7 and 8): "low bias over the NH during April (further analysed in, e.g., Stein et al., 2014), as well as a low bias over Eurasia during August". Need to change "low bias" to "negative bias". In addition, I would say that in August the negative biases occur not only over Eurasia, but over most continental areas of the northern hemisphere.*
*The authors could also be a bit more quantitative and provide spatial correlations between modelled and observed CO for each of the maps on Figures 7 and 8. They could even show maps with biases in a supplement if some of the biases I mention are not so clear.*

As for NO2, we now present observed CO tropospheric columns together with the model biases, to better diagnose the magnitude of respective biases. We do not believe that the spatial correlations provide added value that helps interpretation of the results.
We change 'low bias' to 'negative bias', thank you for spotting this.

*By the way, apart from Stein et al. (2014), the authors could also cite this previous paper regarding the low CO bias over the NH:*
*Shindel et al., Multimodel simulations of carbon monoxide: Comparison with observations and projected near-future changes, 2006, https://doi.org/10.1029/2006JD007100.*

We now include the Shindel et al., reference here.

*\* Lines 563-565: This sentence is a bit awkward: "Considering its strong diurnal cycle, due to the fast photolysis rate, here only daytime values have been used to construct the averages, because the observations from the various field campaigns were equally conducted in daylight conditions".*
*It would be convenient to split all this information into two separate sentences. Moreover, the authors should mention not only the fast photolysis rate but also the strong diurnal cycle of anthropogenic emissions.*

We follow the reviewer suggestion to split the sentence in two. As for the diurnal cycle in emissions: this is not fully straight forward. The current model versions do not yet include such a diurnal cycle for anthropogenic emissions, even though literature suggests ~2x larger emissions during daytime compared to night-time, see, e.g., Miyazaki et al., (2017). Therefore, we choose not to refer explicitly to the diurnal cycle in the emissions, as this may raise confusion at this point. Also the aircraft observations mostly sample more background conditions, less affected by direct emissions.

*\* Lines 578–582: "Figures 16 and 17 evaluate tropospheric NO2 using the OMI satellite observations. The simulations deliver generally appropriate distributions … Another interesting finding is a relatively strong negative bias over ...". It is a bit hard to see the biases in the figures. It would be helpful to show analogous maps with those biases (in a supplement).*

We now present bias maps instead of actual model columns.

*\* Line 600: "Compared to the trace gases previously analysed, Nitric acid is not primary emitted but is purely photochemically formed". Reformulate this sentence, because O3 is not primarily emitted. Also, change "Nitric" to "nitric".*

Thank you for spotting this. we now change the sentence to:

**Compared to several of the trace gases previously analysed, nitric acid is not primary emitted but is purely photochemically formed in the atmosphere.**

*Technical corrections*

All technical corrections have been processed, thank you for spotting them.

*\* line 28: Remove comma in "Other, common".*

*line 56: "correlation between tropospheric composition with ENSO conditions". Change to "relationship between tropospheric composition and ENSO conditions" or "dependence of tropospheric composition on ENSO conditions"?*

*line 57: arctic --> Arctic*

*lines 65-66. Change "and, afterwards, their chemistry modules technically integrated" to ". Afterwards, their chemistry modules were technically integrated".*

*line 85: "which is further expanded" --> "which are further expanded"*

*line 130: I think Lacressonnière et al. 2012 is missing in the reference list.*

*lines 139-140: "and 2 for the troposphere for the aqueous oxidation reaction of sulfur dioxide into sulfuric acid"-->"and 2 for the aqueous oxidation reaction of sulfur dioxide into sulfuric acid in the troposphere"*

*Line 186: Brasseur et al. (1998) is missing in the reference list.*

*line 222: "This has consequences for the description of the various emissions". Do you mean "partitioning" instead of "description"?*

*Line 246: "experiment ID's" --> "experiment IDs"*

*Table 3. Remove "in" from "Specifications of the experiments in evaluated.*

*Line 324: No need comma before "over the NH extra-tropics"*

*Lines 326, 327 and probably other parts of the text: Please change "highest" to "the highest"*

*The expression "associated to" is used three times on page 13 as well as in other parts of the text. I think the use of "associated with" is preferable.*

*Line 367: Need space before parenthesis in "deviation(Jöckel".*

*line 389: compare --> compares*

*"Southern Hemisphere (SH)" is spelled out on line 404, but this should have been done before (around line 393).*

*Figure 9: "Izaa" should be changed to "Izaña" (or "Izana" if the "ñ" is problematic) in one of the panels.*

*Figure 10: "monthly mean surface CO by observations (GAW network) and by the model simulations" --> "monthly mean surface CO as derived from observations (GAW network) and model simulations"*

*line 482: "phae" --> "phase"*

*caption of Figures 12, 13, 14, 18 and 19: "Comparison simulated" --> "Comparison of simulated"*

*page 27: The reference "Aydin et al., 2014" is mentioned twice. I believe this paper is from 2011, as indicated in the reference list.*

*lines 549-550: "the correlation showed acceptable values for all versions (R2 >0.7)". The authors should explicitly refer to the "weighted correlation" rather than just "correlation".*

*lines 562-563: "due to its photochemical balance with nitrogen oxide". Do you refer to NO? If so, please change "nitrogen oxide" to "nitrogen monoxide" or "nitric oxide".*

*line 569: Need comma or semicolon before "hence".*

*Caption of Figure 16: "Tropospheric NO2 from OMI" --> "Tropospheric NO2 columns from OMI"*

*Caption of Figure 17: Remove ", and the model biases with respect to this".*

*line 646: Change "is" to "are" in "the averaged differences for O3 (CO) is"*

*Reference list: Benedetti et al. (2009) is included twice.*

*Reference list: I think that Horowitz et al. (2003) is not mentioned in main text.*

*The main text includes these four references: Madronich and Flocke, 1997; Madronich and Flocke, 1999; Madronich et al., 1989; Madronich, 1987. They are somewhat different in the reference list. Please double-check.*

**References:**

Miyazaki, K., Eskes, H., Sudo, K., Boersma, K. F., Bowman, K., and Kanaya, Y.: Decadal changes in global surface NOx emissions from multi-constituent satellite data assimilation, Atmos. Chem. Phys., 17, 807-837, https://doi.org/10.5194/acp-17-807-2017, 2017.

---

## Author Comment (AC2) · 1 Apr 2019

**Response to Executive Editor comment by Astrid Kerkweg.**

Please find below our response. The editor comment is in italic. Our response is in regular font, and changes to the manuscript are given in bold.

*please note, that if only one model is concerned, the title of a GMD manuscript should state the model name (or its acronym) and a version number. These are always important to know even in the case of an evaluation, as different versions might perform differently for the same evaluation procedure. Therefore please change the title of your manuscript accordingly upon revision; e.g., "Quantifying uncertainties due to chemistry modeling – evaluation of tropospheric composition simulations in the CAMS model (version x.y)"*

Thank you for pointing this out. Note that here we evaluate results from three chemistry model versions in the CAMS model, which so far all follow a different versioning scheme. Therefore it is beyond the scope of the title to provide exact details of the version, but their communality is the IFS version adopted here, which is CY43R1, which is a good proxy for the chemistry version adopted. We will change the title accordingly to:

**"Quantifying uncertainties due to chemistry modeling - evaluation of tropospheric composition simulations in the CAMS model (Cycle 43R1)"**

We now furthermore refer to this cycle in the first sentence of the conclusions section.

*Additionally, please note that evaluation papers also need to include a code availability section, telling the reader how to access the exact code version of the evaluated model or providing profound reasons why the code can not be accessed. Furthermore, please provide reasons, why the data is not freely available.*

The editor is correct that we should be more explicit about code availability. Full data public availability is beyond reach, considering the large volume of data produced for these experiments. Data is fully archived on the ECMWF Archiving system (MARS) and selections will be made available freely to interested readers by contacting the first author. Also we provide details on accessibility of model code. According to this, we now change this section to:

**The source code of the chemistry modules are integrated into ECWMF's IFS code, which is only available subject to a license agreement with ECMWF. The IFS code without modules for assimilation and chemistry can be obtained for educational and academic purposes as part of the openIFS release (https://confluence.ecmwf.int/display/OIFS). A detailed documentation of the IFS code is available from (https://www.ecmwf.int/en/forecasts/documentation-and-support/changes-ecmwf-model/ifs-documentation ). The CB05 chemistry module of IFS was originally developed in the TM5 chemistry transport model. Readers interested in the TM5 code can contact the TM5 developers (http://tm5.sourceforge.net). The BASCOE stratospheric chemistry module can be freely obtained from the BASCOE developers (http://bascoe.oma.be). The MOCAGE chemistry module of IFS is developed at Météo-France on the basis of the**

MOCAGE chemistry-transport model, http://www.umr-cnrm.fr/spip.php?article128. The MOZART code can be obtained through contacting their developers via https://www2.acom.ucar.edu/gcm/mozart. The MOZART and CB05BASCOE chemistry schemes are also freely available through the Sander et al. (2019) publication.

The model simulation datasets used in this work are archived on ECMWF archiving system (MARS) under the respective experiment IDs listed in Table 3. Readers with no access to this system can freely obtain these datasets from the corresponding author upon request.

---

## Author Comment (AC3) · 1 Apr 2019

**Response to the second Reviewer**

We thank the referee for his/her positive review and for the provision of useful comments and suggestions. Below we answer them to our best ability. The reviewer comments are in italic. Our responses are in regular font, and changes to the manuscript are given in bold.

This manuscript evaluates tropospheric composition simulations in the CAMS modelling system and quantifies uncertainties related to different chemical schemes. It is well structured and written and illustrates original and interesting results for the . CAMS modelling system. I suggest acceptance of the manuscript for publication after taking into consideration the following comments.

**Main Comments**
*1) I guess that the simulations were carried for the year 2011 but I think the authors should describe in Section 2.3 which was the time period that the simulations were carried out.*

Model simulations for 1 July 2010 to 1 January 2012 have been carried out. We now include such a statement explicitly:

To allow for sufficient model spinup, **the model versions are initialized for 1 July 2010 and ran through until 1 January 2012**.
**The first 6 months of the simulation are considered as spin-up and therefore not evaluated.**

*2) The authors mention that the averaging of large number of measurements over space and time partly solves the problem of interannual variability (lines 273-275 in page 11). Can this dataset of Emmons et al. (2000) be representative to compare with the CAMS simulations for the year 2011? I understand the uniqueness of this dataset but could the authors clarify this issue and discuss the uncertainties and the weaknesses of this comparison?*

Indeed the referee is fully correct with his analysis. It is also true that for the *total* anthropogenic VOCs emissions, the changes between the year 1990 and 2011 are of the order of 14%, following Emissions Database for Global Atmospheric Research (EDGARv4.3.2 database). Therefore obviously caution has to be taken when analyzing the comparison, but we believe the aircraft dataset comparison to be still a valid methodology despite the large temporal difference between observations and modelled data because of the following two reasons:
We expect the impact of this change to be lower at background locations or outflow regions, as included in the comparison of presented in this manuscript, only partly affected by anthropogenic emissions, while biogenic emissions are expected to remain largely unchanged. Also the variability and measurements uncertainties present in the observations are larger than 14%, implying that we can still consider these observations representative, especially because they are averages over larger regions in space and time. To make this clear, we now write in the manuscript:

**For the total anthropogenic VOCs emissions the changes between the year 1990 and 2011 are of the order of 14%, following Emissions Database for Global Atmospheric Research (EDGARv4.3.2 database). Nevertheless, the evaluations presented here are all sampling background locations or outflow regions, and are hence only partly affected by such changes in anthropogenic emissions. Also the variability as well as measurement**

**uncertainties present in the observations are larger than 14%, implying that we can still consider these observations representative.**

*3) I would suggest the authors to provide a short description of the method used to calculate the weighted values of bias and correlation in Table 4.*

The method has been extensively described in Jöckel et al. (2006), where the mathematical derivation is also explained in the appendix. As the mathematical description of the method would be too tedious, we have added the following short description to the manuscript, referring to Jöckel et al (2006) for detailed information. We now write

**As explained in further detail by Jöckel et al. (2006), with this approach, the measurement locations with high variability have less weight, whereas more weight is given to stable, homogeneous conditions. This allows us to compare values that are more representative for the average conditions and to eliminate specific episodes that cannot be expected to be reproduced by the model.**

*4) The authors write in line 375 (page 15) that " CBA is the only model version to deliver a satisfactory bias" . Is this a robust conclusion? What is statistically satisfactory? Looking Table 4 I see that in some species CBA bias is smaller than in other schemes, in some other species the biases are comparable and in other species the CBA bias is worse.*

Note that this specific comment on this line only referred to $SO_2$, so the reviewer is correct that biases in CBA are not always the best. In general, whether a bias is *satisfactory* small will depend on the application area, which is indeed not detailed in present work. Instead, in Sec. 5 we defined a weighted bias, which relates the bias to the standard deviations in the model and observations. This value should be between [-1,1] to deliver satisfactory results.
Therefore we choose to re-formulate the specific sentence here to:

**For $SO_2$ CBA is the only model version to deliver a weighted bias that is larger than -1.**

*5) In the evaluation of ozone the authors conclude that "overall, the evaluation at individual station provides reasonable agreement between model simulations and sondes". How these evaluation results compare with other ozone evaluation studies which were based on MACC and CAMS products (e.g. Inness et al., 2015; Katragkou et al., 2015; Akritidis et al., 2018). I think this conclusion could be also supported by these studies.*

So far we didn't show evaluation results from other CAMS products as this is beyond the scope of current manuscript. Furthermore, one should realize that chemistry versions and model configurations as adopted here are to some extent different compared to those use in important MACC/CAMS products such as the reanalyses.
However, to aid the interpretation of the model quality, i.e. to put the current model performance into perspective, we now include an assessment of the CAMS Interim Reanalysis (CAMSiRA, Flemming et al., 2017) in the evaluation against ozone sondes. We choose only to show figures for the annual average, zonal average bias and RMSE at various altitude ranges, to give a general indication of our model performance relative to that of CAMSiRA. Indeed, this evaluation shows that biases and RMSE are within the range of those of CAMSiRA, with the free running model versions of equal (or better) performance towards the boundary layer, and CAMSiRA generally better in the free troposphere. For further details about the configuration

and performance the reader is referred to Flemming et al. (2017) and Inness et al. (2019). We now write:

**In this evaluation we also present data from the CAMS Interim Reanalysis (CAMSiRA) for the year 2011, to put the current model evaluation into perspective.** This summary analysis shows averaged biases within ±10 ppbv, which is also in line with the $O_3$ bias statistics against the aircraft climatology. **At lower altitudes the model biases are mostly equal or better than those from CAMSiRA, while above 500 hPa CAMSiRA delivers mostly smaller biases thanks to the assimilation of satellite ozone observations.** The RMSE shows a larger spread in the lower troposphere of the NH, while at higher altitudes, above 500 hPa the overall magnitude of the RMSE for the three chemistry versions converges to values ranging from 10 to 16 ppbv, depending on the latitude. **Here the CAMSiRA shows overall better performance, mainly for the tropics and SH, while over the NH its performance is similar to IFS(CBA).**

*6) In lines 448-449 (page 20) the authors write "Approximately half of the CO burden is directly emitted, and the rest formed through degradation of methane and other VOC's". Please add a relevant reference.*

We now add Hooghiemstra et al. (ACP 2010) as reference for this statement, as they provide a detailed evaluation of a-priori and optimized budgets for global CO production.

*7) On how many data points (and years) the temporal correlations shown in Figure 10 are based?*

The temporal correlation presented in Figure 10 is based on twelve points (the monthly means) per station, and was evaluated for the year 2011. Therefore this figure shows an evaluation of the model ability to represent the seasonal cycle, as discussed in the manuscript.

*8) In lines 526-528 (page 26) it is written "The vertical profiles (see Figure 13) are strongly biased (e.g., SONEX, Newfoundland and PEM-Tropics-A, Tahiti), with positive biases occurring at the surface and negative in the free troposphere."*
*Could this result also related to inadequate outflow from the atmospheric boundary layer (ABL) to the free troposphere (FT) and hence to model weakness in ABL-FT exchange?*

Thank you for this interesting suggestion. Although such processes of vertical mixing could certainly contribute to uncertainties in the vertical distribution of C2H4, this would then also affect any of the other chemical tracers, as well as meteorological variables (e.g. humidity, temperature), which generally do not show indication of this type of issues. Therefore so far we have no indication that such an uncertainty is driving the discrepancy in the modeled vs observed $C_2H_4$ profiles, but rather believe that our emissions and chemistry contain larger uncertainties, as currently stated on the manuscript.

*9) The authors refer to correlation R (that span from -1 to 1) but showing R2 which practically describes the explained variance. Although this is not crucial in the discussion it could propagate a misunderstanding on these statistical parameters when the article is published. I would suggest to modify this accordingly.*

The reviewer is correct that there can be some confusion between the use of correlation in terms of R and $R^2$. Throughout the text we make sure to refer to $R^2$ when providing quantitative reference to the correlation, and at the start of Sec. 5 we now explicitly write:

"Table 4 summarizes the comparison of the various model results with aircraft measurements, described in Sec. 3.1, in terms of biases and correlation, **in terms of explained variance ($R^2$)**, …"

While or the table header we now write:

"Table 4. Summary of the Bias and correlation coefficients **(in terms of explained variance, $R^2$)** …"

*10) Generally, I think that the discussion in model difference is rather technical and I would suggest the authors to discuss also the possible scientific reasons for discrepancies among the simulations with different chemical schemes for the different chemical species.*

Indeed this manuscript has a largely technical focus, explaining the current state of the modeling system, its validation, and its general ability to quantify uncertainties due to model chemistry. Any more scientific reasons to explain discrepancies among simulations inherently require additional sensitivity studies, which is beyond the scope of this work. Having said this, we now do pay more attention to differences in model performance related to differences in their configurations, as also requested by the other reviewer, see particularly our responses to his ´Main Comments´. We hope this addresses the concerns raised here.

**Minor comments**
*page 13, line 348: should rather be "relative shorter" instead of "relative short"*

Thank you, we changed this accordingly.